# Hydrodynamics of pedestrians' instability in floodwaters

Chiara Arrighi[1], Hocine Oumeraci[2], Fabio Castelli[1]

[1]Department of Civil and Environmental Engineering, University of Florence, Via di S. Marta 3, 50139, Firenze, Italy
[2]Leicthweiss Institute for Hydraulic Engineering and Water Resources, TU Braunschweig, Beethovenstrasse 51a, D38106, Braunschweig, Germany.

Correspondence to: Chiara Arrighi (chiara.arrighi@dicea.unifi.it)

**Abstract.** People safety is the first objective to be fulfilled by flood risk mitigation measures and according to existing reports
on the causes of casualties, most of the fatalities is due to inappropriate behaviors like walking or driving in floodwaters. Currently available experimental data on people instability in floodwaters suffer of a large scatter primarily depending on the large variability of physical characteristics of the subjects. This paper introduces a dimensionless mobility parameter $\theta_P$ for people partly immersed in flood flows, which accounts for both flood and subject characteristics. The parameter $\theta_P$ is capable of identifying a unique threshold of instability depending on Froude number thus reducing the scatter of existing experimental
data. Moreover, a 3D numerical model describing a detailed geometry of a human body and reproducing a selection of critical pairs of water depth and velocity is presented. The numerical results in terms of hydrodynamic forces and force coefficients are analyzed and discussed. Both the mobility parameter $\theta_P$ and the numerical results hint the crucial role of Froude number and relative submergence as the most relevant dimensionless numbers to interpret the loss of stability. Finally, the mobility parameter $\theta_P$ is compared with an analogous dimensionless parameter for vehicles instability in floodwaters, providing a new
contribution to support flood risk management and people education.

**Keywords: flood risk, dimensional analysis, Froude number, human-water interaction**

## 1 Introduction

Floods are among the main natural disasters in terms of deadliest events and economic damages (Munich Re, 2015b). The
2011 flood in Thailand caused 40 billion dollars of overall damages (Munich RE, 2012), the 2014 flood affecting India and Pakistan caused 665 fatalities (EM-DAT, 2012; Munich Re, 2015a). Although the number of fatalities caused by floods is lower than other hazards (i.e. earthquakes), flood events are those affecting the largest number of people (EM-DAT, 2012). Among the possible human interactions with the hydrological cycle, the loss of life in case of inundation represents a crucial phenomenon where flood and subjects characteristics dynamically interact. The loss of stability of a human body in floods,
can be seen as the most direct, tangible and fastest interaction between water and a human system. This topic has been less

studied in recent years with respect to other branches of socio-hydrology, which well conceptualized long-term human-water interactions (Di Baldassarre et al., 2013a; Di Baldassarre et al., 2013b; Di Baldassarre et al., 2015).

Research on loss of life in floods is sparse, and has been so far focusing on dam break catastrophes (Aboelata and Bowles, 2008; Chakraborty et al. 2005), physical experiments (Abt et al. 1989; Karvonen et al. 2000; Jonkman and Penning-Rowsell, 2008; Xia et al. 2014) and conceptual models (Love, 1987; Lind et al., 2004; Milanesi et al., 2015). Most of loss of life models are based on the location of the population at risk measured by its distance from the dam and account for the depth of flooding, population distribution, and effectiveness of warning and evacuation processes (Jonkman et al., 2002; Aboelata and Bowles 2008; US Department of Homeland Security, 2011; Penning-Rowsell et al., 2005). Other models relate the mortality to past flood events (Brown and Graham,1988), which may not be representative anymore of current situation (Jonkman et al. 2002). In the last decade, two approaches to flood fatalities assessment, namely individual and societal risk, have been identified (Tapsell et al., 2002; Beckers et al., 2012; de Bruijn et al. 2014).

The characteristics of the flood and floodwater along with the characteristics and behaviour of the population determine the likelihood of a death due to flooding (Di Mauro et al. 2012). It is widely recognized that, in developed countries, the majority of flood-related fatalities occur as a result of inexperienced people entering floodwater either in boats, vehicles or on foot (Franklin et al. 2014). Many studies (Jonkman and Kelman, 2005; Maples and Tiefenbacher, 2009; Fitzgerald et al. 2010; Kellar, 2010) have shown that the first cause of death during a flood event is related to roads and vehicles (Arrighi et al., 2015). Jonkman and Kelman (2005) reported that in the Netherlands the 33% of deaths for drowning occurs in a vehicle and the 25% as a pedestrian. Many casualties in fact, occur when people try to move in floodwaters (Di Mauro et al. 2012; Chanson et al. 2014), in this case previous experiences may play a role (Siegrist and Gutscher, 2008). Thus, understanding the instability mechanisms and identifying the safest behaviour when a pedestrian or vehicle is unexpectedly facing a flood, might be of crucial importance for management strategies (Franklin et al. 2014; Di Mauro et al. 2012) and emergency planning (Simonovic and Ahmad, 2005).

Two hydrodynamic mechanisms that can cause human instability have been distinguished in existing studies: moment instability (toppling) and friction instability (sliding). Toppling occurs when the mobilising moment caused by the incident flow exceeds the resisting moment caused by the resultant weight of the body (Abt et al., 1989; Jonkman and Penning-Rowsell, 2008). Sliding occurs if the drag force induced by the flow is larger than the frictional resistance between the person's feet and the substrate surface (Keller and Mitsch, 1993).

Foster and Cox (1973) tested the instability perception of children with different physical characteristics (i.e. height and mass combinations) in a laboratory flume and found that also physical, emotional and dynamic factors deeply affect human stability under water flow. They observed that sliding instability prevailed, since the tests were performed with high flow velocities and low water depths. Further tests by Abt et al. (1989) showed that toppling instability is crucial for higher water depths. More recently, several laboratory tests were performed on real adults and children (Takahashi et al., 1992; Keller and Mitsch, 1993; Karvonen et al. 2000; Yee, 2003) considering different training, wearing, environmental conditions and definitions of instability. These studies provide an extensive dataset, which has been used to define inversely proportional

linear relationships between mean flow velocity and depth (Cox et al., 2010; Smith, 2015), which are often adopted as a reference for flood hazard zoning. These empirical approximating functions are however purely regressive and do not allow linking hazard levels and physical effects. Jonkman and Penning-Rowsell (2008) extended the existing experimental data by testing an adult stuntman in real channel conditions of low depth and significant velocity. Moreover, they calibrated a

simplified model for adults which accounts for both slipping and toppling using the data by Abt et al. (1989) and Karvonen et al. (2000). The buoyancy force is however neglected.

Recently, Xia et al. (2014) carried out experiments on a rigid human body model with geometric scale 1:5.54. They developed a parametric scheme, introducing buoyancy force and considering both toppling and slipping failure mechanisms. They derived two formulae for the critical velocity for slipping and toppling instability mechanism.

Conceptual models were introduced to describe the human stability as a function of flow velocity and water depth in order to provide an interpretative framework for the experimental activities. These models are based on different assumptions regarding the shape of the body, the involved forces and the failure mechanisms. Love (1987) modelled the human body as a rectangular monolith and recognized the role of the buoyancy force on toppling instability. Lind et al. (2004) tested both conceptual and empirical formulae and calibrated a relation based on the concept of the depth-speed product number (i.e. water depth

multiplied by flow velocity). They modelled the human body as a rigid circular cylinder and proposed an equation for toppling instability, which yields the critical depth speed product number as a function of drag coefficient and submergence. Walder et al. (2006), studying a tsunami induced by a debris flow, developed a simplified approach to predict critical velocity for slipping to occur, disregarding toppling instability and the role of the buoyancy force and assuming a fixed drag coefficient. They also supposed that, in waters of some sufficient depth, people could not stand even if the flow velocity is negligible. Milanesi et

al., (2015) recently introduced a conceptual model for people instability in a fluid flow also considering the effects of the local bottom slope and the density of the fluid.

Over the last four decades, a number of laboratory-based experimental studies have been undertaken to define the limits of stability under different flow regimes. Moreover, different conceptual models have been developed to derive formulae for these stability limits, usually assuming fixed values for the drag coefficient. The very wide scatter of critical pairs of water

depth and velocity is the main evidence of the existing experiments on people instability. A large scatter exists within the same dataset and to a more significant degree, when all datasets are combined (Cox et al., 2010, Russo et al., 2013). In fact, instability conditions are strongly affected by diverse 'non-hydraulic' parameters (Martinez-Gomariz et al., 2016), including the physical characteristics of the subjects (i.e. weight and height), their level of training, clothing and experimental conditions. Thus, a synthetic identification of hazard regimes in dimensional terms is quite difficult.

The aim of this work is to overcome the scatter of existing experiments on people instability under water flow introducing a dimensionless criterion capable of accounting for both flood and human characteristics and to understand its dependency on flow regime (Sect. 2). The dimensional analysis is undertaken to clarify if a limited number of parameters (i.e. related to hydraulics and body shape) is capable of explaining most of the variability observed in the experiments or if non-hydraulic parameters are still relevant. The dimensionless criterion also allows comparing the instability conditions of people with the

incipient motion conditions of vehicles (Arrighi et al., 2015), thus providing a framework for societal risk assessment, risk management and education. Particularly for educational purposes, the use of dimensionless quantities may favour the definition of safety rules, e.g. recognizing a level of submergence (water depth reaching knees, ankles or waist) is easier than referring to absolute water depths. Since examples of numerical model can be hardly found in literature, in order to better understand the hydrodynamic interaction between human body and mean flood flow, a simplified 3D numerical model, describing a detailed human geometry partly immersed in water, is introduced (Sect. 3) and used to reproduce a selection of the existing experimental data available in literature.

## 2 Instability conditions of a human body under water flow

### 2.1 Geometric representation of the human body and acting forces

The shape of the human body is extremely complex, thus leading to different conceptualization schemes of its geometry, such as prisms or cylinders (Abt et al., 1989; Lind et al., 2004; Milanesi et al., 2015). Moreover, relative motion, posture, clothing and physical parameters (i.e. body type, size and build) influence the hydrodynamic interaction between human body and water flow. The two mechanisms by which the stability of people is lost in floodwaters are sliding and toppling (Abt et al., 1989; Lind et al., 2004; Jonkman and Penning-Rowsell, 2008). Incipient sliding on a horizontal bed occurs when drag force on the human body just exceeds the friction force of the feet on the bottom, while motion due to toppling occurs when the moment exerted by drag force just exceeds the weight-induced moment. For a stationary body in a moving flow lift force and drag force are the force components acting normal and parallel to the mean direction of the undisturbed flow respectively (Vickery, 1966).

In order to minimize the number of parameters, the shape of the human body is mechanically schematized as in Lind et al. (2004), with reference to an approximating prism of height $H_p$, width $l$ and length $d$ (Figure 1, panel a). In the inset of Fig. 1 (panel a), $W$ is the weight, $B$ is the buoyancy, $Li$ is the lift force and $D$ is the drag force. $H$ is the water depth and it is assumed that the resultant drag force acts on one-half the water depth $H/2$ (Lind et al., 2004). The lever arm of the stabilizing force, which is the weight minus buoyancy and lift effect, is $d$, which is assumed as commutable with the full length of the foot. For a uniform prism, the lever arm of the stabilizing moment should be equal to $d/2$, but here it is preferred to use the length of the foot $d$ in order to partly accounting for the natural adjustment of the posture of the subject observed in the experiments. In the panel (a) in Fig. 1, the rotation point O is placed on the toe for average flow velocity $U$ coming from right to left. Otherwise, for a flow velocity oriented from left to right, the rotation point O would be placed on the heel.

### 2.2 Dimensionless mobility parameters

The definition of the dimensionless mobility parameter for people instability under water flow follows the procedure adopted for the introduction of the mobility parameter for vehicles incipient motion as in Arrighi et al. (2015). It starts with defining the forces acting on the body and then proceeds with the separation of dynamic and static actions in order to identify relevant

dimensionless groups of variables. The two mechanisms by which the stability of people is lost in dynamic conditions in floodwaters (i.e. sliding and toppling) are separately analysed. For hydrostatic conditions (i.e. zero flow velocity), the hydrodynamic actions are null and the static equilibrium is obtained equalling weight and buoyancy force. Sliding equilibrium is considered first. Incipient sliding on a horizontal bed occurs when drag force D on the human body just exceeds the friction force of the feet on the bottom. Referring to Fig. 1 (panel a), the friction force is equal to the effective weight (weight $W$ minus buoyancy $B$ and lift force $Li$) multiplied by the static friction coefficient $\mu$. The sliding instability condition is then

$$D > (W - B - Li) \cdot \mu \tag{1}$$

The weight $W$ is the product of constant human body density $\rho_p$, acceleration of gravity $g$ and body volume $H_P \cdot d \cdot l$. The width of the prism $l$ (Fig. 1, panel b) assumed equal to the waist diameter, has been graphically found as a good proxy for the average width of the human body for a mesomorphic individual (Beashel and Taylor, 1997)

$$W = \rho_P \cdot g \cdot \left( H_P \cdot d \cdot l \right) \tag{2}$$

Buoyancy force $B$ is the product of water density $\rho$, acceleration of gravity $g$ and the immersed volume of the prism

$$B = \rho \cdot g \cdot \left( H \cdot d \cdot l \right) \tag{3}$$

Drag and lift forces are a function of the square of flow velocity U and are referred to the same total frontal area of the prism, normally projected to the flow $H_P \cdot l$. This reference area has been preferred to the wetted area, because the determination of the actual wet area requires the study of the water profile due to the flow-body interaction, i.e. the wetting water depth does not coincide with the undisturbed water depth $H$ (see also section 3.2). In fact, as shown in Fig. 3 the difference between actual and undisturbed water depth is not negligible for supercritical flows.

$$D = \frac{1}{2} \cdot \rho \cdot U^2 \cdot C_D \cdot H_P \cdot l \tag{4}$$

$$Li = \frac{1}{2} \cdot \rho \cdot U^2 \cdot C_l \cdot H_P \cdot l \tag{5}$$

where $C_D$ and $C_l$ are the drag and lift coefficient respectively. As shown by Arslan et al. (2013), Zhang et al. (2014) and by Arrighi et al. (2015) lift force can play a significant role also for partly submerged objects. The average density of the human body ($\rho_P = 1062$ kg/m³) is generally assumed equal to the density of muddy water, thus $\rho_p$ is substituted with $\rho$ in Eq. (2). The assumption $\rho_p = \rho$ implies that a human body immersed in water can experience a condition of static equilibrium. Substituting the Eqs. (2), (3), (4), (5) in Eq. (1) and putting equal the left and right term to define the equilibrium condition the following equation is obtained

$$\frac{1}{2} \cdot \rho \cdot U^2 \cdot C_D \cdot H_P \cdot l = \left( \rho \cdot g \cdot \left( H_P \cdot d \cdot l \right) - \rho \cdot g \cdot \left( H \cdot d \cdot l \right) - \frac{1}{2} \cdot \rho \cdot U^2 \cdot C_l \cdot H_P \cdot l \right) \cdot \mu \qquad (6)$$

The variables $l$ and $\rho$ multiply all the terms of Eq. (6), thus they can be simplified. Separating the dynamic terms ($\propto U^2$) from the static terms Eq. (6) is simplified as

$$\frac{1}{2} \cdot U^2 \cdot C_D \cdot H_P + \left( \frac{1}{2} \cdot U^2 \cdot C_l \cdot H_P \right) \cdot \mu = \left[ \left( g \cdot H_P \cdot d \right) - \left( g \cdot H \cdot d \right) \right] \cdot \mu \qquad (7)$$

5   Collecting $U^2$ in the left term and d in the right term, then dividing both terms for $1/2 \cdot \mu \cdot H \cdot g$ the equilibrium condition yields

$$\frac{U^2}{gH} \cdot \left( C_l + \frac{C_D}{\mu} \right) = \frac{2d}{H_P} \cdot \frac{H_P - H}{H} \qquad (8)$$

Where

$$Fr^2 = \frac{U^2}{gH} \qquad (9)$$

is the square of the Froude number of the undisturbed flow,

$$C_s = \frac{C_D}{\mu} + C_l \qquad (10)$$

and $C_s$ includes the coefficients for drag $C_D$, for lift $C_l$ and for friction $\mu$ forces.

$$\theta_P = \frac{2d}{H_P} \cdot \frac{H_p - H}{H} \qquad (11)$$

is defined as the dimensionless mobility parameter for sliding instability of people standing in floodwaters. $\theta_P$ is composed by two factors: the shape factor $2d/H_P$ and the relative dry surface of the body $(H_P-H)/H$. $\theta_P$ depends on Froude number and on
15   the dimensionless force coefficients. If the assumption $\rho_p = \rho$ is removed, a factor $\rho_p/\rho$ is introduced in Eq. (11), which turns into the more general form of Eq. (12)

$$\theta_P = \frac{2d}{H_P} \cdot \frac{\dfrac{\rho_p}{\rho} H_p - H}{H} \qquad (12)$$

It should be noticed that with the general form of Eq. (12) the height of the subject appears virtually increased of about the 6% which corresponds to an increased stability of the subject. Moreover, the water density $\rho$ varies with temperature and
20   concentration of dissolved compounds and suspended load, thus Eq. (12) can be used to account for any fluid/body density. Toppling instability occurs when the moment induced by drag force around a pivot point (i.e. the heel or toe) just exceeds the moment of the resultant vertical force (body weight $W$ minus buoyancy $B$ and lift force $Li$) as shown in Fig. 1 (panel a)

$$(W - B - Li) \cdot d = D \cdot \frac{H}{2} \quad (13)$$

Substituting the forces $W$, $B$, $Li$ and $D$ (Eqs. 2, 3, 4, 5) in Eq. (13), the following threshold condition for incipient toppling is obtained

$$\left[ (\rho \cdot g \cdot l \cdot H_P \cdot d) - (\rho \cdot g \cdot l \cdot H \cdot d) - \frac{1}{2} \rho \cdot U^2 C_l \cdot H_P \cdot l \right] \cdot d = \left( \frac{1}{2} \cdot \rho \cdot U^2 C_D \cdot H_P \cdot l \right) \cdot \frac{H}{2} \quad (14)$$

$\rho$ and $l$ can be dropped on both side of Eq. (14), which after some manipulations and simplifications yields:

$$\frac{U^2}{gH} \cdot \left( \frac{H}{2d} \cdot C_D + C_l \right) = \frac{2d}{H_P} \cdot \frac{H_P - H}{H} \quad (15)$$

This represents a relationship between the square of Froude number $\dfrac{U^2}{gH}$ together with the dimensionless parameter $C_t$

$$C_t = \left( \frac{H}{2d} \cdot C_D + C_l \right) \quad (16)$$

on the left hand side, and on the right hand side a mobility parameter for toppling instability conditions for a person in
floodwaters $\theta_{Pt}$

$$\theta_{Pt} = \frac{2d}{H_P} \cdot \frac{H_P - H}{H} \quad (17)$$

The mobility parameter $\theta_{Pt}$ obtained for toppling is equal to the mobility parameter $\theta_P$ introduced for sliding (Eq. 11). However, different is the combination of coefficients $C_s$ (Eq. 10) and $C_t$ (Eq. 16) that define the instability limit. In fact, for toppling instability conditions, $C_D$ is multiplied by $H/2d$ in Eq. (16), which can be interpreted as a measure of the relevance of the
moment induced by the drag force for larger water depths H.  Therefore, although there are two different incipient motion mechanisms, a unique parameter $\theta_P$ accounting for a limited number of human body parameters ($H_P$ and $d$) and flow characteristics is able to represent both mechanisms. $\theta_P$ is meaningful for $0<H<H_P$. From the physical point of view the limit H=0 corresponds to extremely high Fr tending to infinity, H=$H_P$ corresponds to fully submerged condition where static equilibrium occurs given the assumption $\rho_P=\rho$.  It should be noticed that the mobility parameter for people $\theta_P$ could be also
obtained from the mobility parameter for vehicles $\theta_V$, defined by Arrighi et al., (2015) considering a human body as a 'special' vehicle model with elevation of the planform $h_c$ equal to zero, length equal to $d$ and density $\rho_c$ equal to water density $\rho$.

For the general equation of the mobility parameter $\theta_P$ (Eq. 12), the sensitivity has been evaluated with respect to length of the foot $d$, height of the subject $H_p$ and human body density $\rho_P$. The sensitivity is assessed with a local method, i.e. calculating the partial derivative of $\theta_P$ with respect to the selected factors $X_j$. The analytical formulas allows calculating the sensitivity for
each value of the water depth H thus identifying possible critical ranges. Table 1 shows the sensitivity functions with respect to the selected parameters. The units of measurement of the sensitivity function are length$^{-1}$ and length$^3$/mass for the geometric

parameters and density parameter respectively. The sensitivities of $\theta_P$ to $\rho_P$ and $d$ decrease with increasing water depth $H$. The sensitivity to $\rho_P$ and $d$ are of the order of magnitude of $10^{-3}$ and $10^{-1}-10^1$ respectively, for the experimental range of water depths. This means that small variations of $\rho_P$ are negligible for $\theta_P$, thus the assumption $\rho_p=\rho$ is not affecting significantly the results. High sensitivity to $d$ is found particularly for water depths lower than 0.5 m. Thus, d is a more sensitive parameter,

although its variation is physically constrained because the foot to height ratio is in the range 0.149-0.169 according to allometry studies (Davis, 1990; Pawar and Dadhich, 2012; Fessler et al., 2004). The sensitivity to $Hp$ is of the order of magnitude of $10^{-1}$ for height of the subjects between 1 and 2 m (i.e. children and adults).

Therefore, since the sensitivity of the parameter $\theta_P$ is the product of the sensitivity function and the variation of the parameter, $\theta_P$ is robust enough, although obviously its regression function depends on the experimental data used (see Fig.1).

The dimensionless mobility parameter $\theta_p$ indicates that the stability of a human body in floodwaters is related to relative submergence and Froude number. The mass does not appear in the parameter definition because with the dimensional analysis the mass becomes a density $\rho_p$. All human subjects tested in the experiments had different mass/weight but had the same density and the dimensional analysis allows identifying dimensionless combinations of the variables of the system for a given set of independent fundamental units. Also if we do not assume $\rho_p=\rho$ (Eq. 12) we obtain a constant factor 1.062, which virtually

increases the height. The height in fact can be seen as a sort of 'proxy' of the weight for a mesomorphic individual since the mass is the product of body density and body volume (and the body volume depends on the height of the subject).

The mobility parameter $\theta_P$ is introduced for null bed slope and density of floodwater coinciding with the density of clear water to strictly follow the experimental set-up of the selected studies (Sect. 2.3), however, a further study could also modify Eq. (17) to account for any terrain slope and water density.

**2.3 Dimensionless instability threshold from Foster and Cox (1973), Karvonen et al. (2000), Jonkman and Penning-Rowsell (2008) and Xia et al. (2014) flume experiments.**

A selection of the existing flume experiments on people instability in flood flows has been made to test the applicability of the mobility parameter $\theta_P$. This selection covers a wide range of Froude numbers and accounts for different subjects characteristics and for a human scale model. The available datasets (Foster and Cox 1973, Karvonen et al. 2000, Jonkman and Penning-

Rowsell 2008 and Xia et al. 2014) provide the experimental pairs of water depth and velocity (H, U) in which the subjects lose their stability together with subjects' physical characteristics (i.e. weight and height). The length of the foot $d$ is calculated as a fraction of the height, which is a standard assumption in human allometry (Davis, 1990; Pawar and Dadhich, 2012; Fessler et al., 2004). The dataset by Abt et al. (1989) and by Takahashi et al. (1992) have not been included because in the first case the experimental conditions are considered not fully representative of a 3D flow and limited in the investigated range of flow

regimes, in the latter it was not possible to retrieve the heights of the tested subjects to calculate the mobility parameter.

A diagram showing the mobility parameter $\theta_P$ against Froude number for the selected experimental data is drawn in Fig. 1. The mobility parameter $\theta_P$ evaluated for experimental pairs (*H, U*) (Foster and Cox 1973, Karvonen et al. 2000, Jonkman and

Penning-Rowsell 2008 and Xia et al. 2014) defines in the diagram in Fig. 1 a unique dimensionless critical threshold of instability $\theta_{Pcr}$ for people under water flow that can be approximated as

$$\theta_{Pcr} = 1.57 \cdot Fr + 0.057 \qquad (18)$$

The determination coefficient $R^2$ and root mean square error RMSE of the regression curve are 0.98 and 0.21 respectively.

Since the regressed critical threshold curve is linear it may appear inconsistent with Eq. (8) or (15) where $\theta_P$ depends on the square of Froude number. The 3D numerical model described in Sect. 3 and the numerical results of the parameter study (Sect. 4) will help clarifying this apparent inconsistency demonstrating the dependency of $C_s$ (Eq. 10) and $C_t$ (Eq. 16) on the inverse of Froude number.

The critical mobility parameter $\theta_{Pcr}$ ranges from 0.3 for low Froude numbers (i.e. sub-critical conditions) up to 6 for super-
critical flows and identifies a threshold, which separates stable conditions above the curve from unstable conditions below the curve, with no discontinuity between the two motion mechanisms. While the datasets by Karvonen et al. (2000) and by Jonkman and Penning-Rowsell (2008), represented with circles and diamond symbols respectively, are well aligned, the datasets by Foster and Cox (1973) and by Xia et al., (2014) appear to be more scattered. The first two datasets refer to adult subjects with different age, weight and height, the third refers to children and the latter to the human scale model. Particularly,
the selection of points calculated from the data by Xia et al. (2014) are above the threshold curve, i.e. they lie in the stable side of the diagram. This confirms that the experimental instability conditions obtained for a human scale model are more conservative than critical condition for human subjects as argued by the authors (Xia et al., 2014). Some of the data by Foster and Cox (1973) instead are under the curve, thus in the unstable portion of the diagram. A lower estimation of the mobility parameter may be due to the values assigned to the foot length. In fact, the growth of the feet is not proportional to the growth in height for children in the development age and common foot to height ratios are valid for adults (Davis, 1990; Pawar and
Dadhich, 2012; Fessler et al., 2004).

The mobility parameter $\theta_P$ demonstrates that a reduction in the scatter of the existing instability diagrams is possible if the analysis of the instability threshold is done in dimensionless terms and accounts for both flood and subject characteristics. Moreover, a dependence of $\theta_P$ on the dimensionless force and friction coefficients has been found (Eqs. 10, 17). The analysis
of the force coefficients requires a separate and dedicated analysis through a numerical model, which might help clarifying the hydrodynamics of instability mechanisms (Sect. 3).

## 3 Numerical model

### 3.1 Model description

The main aim of the numerical simulations is to understand how different mean flow regimes, in which people instability is
experimentally observed, affect the drag and lift forces and the motion mechanisms. Thus, the focus is to assess the physical dependencies among the involved parameters (i.e. force coefficients and Froude number) and relate them to the mean flow

properties. The study is focused on the estimation of integral quantities such as forces, rather than on the detailed description of the flow properties in terms of local distributions. Thus, the numerical simulations were performed using the 'laminar' turbulence settings of the numerical code, avoiding the calibration of the turbulence model coefficients, which however could not have been possible with the existing data. Laminar settings of the code do not force a laminar flow simulation, which

would not be physically consistent, but they simply refer to the absence of turbulence modelling. A turbulence model was not selected for two main reasons: first, a turbulence model needs the calibration and/or validation of some coefficients and existing experiments were not available for this purpose; second, a rigid body approximates the experimental conditions for subjects allowed to move freely. This obviously bears an error in the estimation of the forces on the subject, thus this simplifying assumption was considered adequate for the intrinsic uncertainties of the simulated problems. As in Arrighi et al. (2015)

preliminary tests have shown the substantial independence of the results on the particular choice of the closure model for the selected mesh size. Moreover, the model adequately reproduced the flow around a circular cylinder used as a benchmark test, with a correct estimation of pressures and drag coefficients for the selected range of Reynolds number.

For the numerical simulations, the CFD toolbox OpenFOAM® (www.openfoam.com) is used since it is proven suitable for numerical modelling of wide number of applications in coastal and hydraulics engineering (Leclercq and Doolan, 2009; Seo

et al., 2010; Arrighi et al., 2015). The code includes several tools and utilities for wave/current generation/absorption, mesh manipulation and turbulence modelling. The solver waveFoam included within the library waves2Foam (Jacobsen et al., 2012) is selected because it handles two incompressible, isothermal, immiscible fluids with capturing of the fluid-fluid interface through the volume of fluid (VOF) method. It solves the Reynolds Averaged Navier Stokes (RANS) equations implemented in OpenFOAM and applies the relaxation zone technique for current generation together with absorption of its reflection. This

'active sponge' layer is a practical boundary condition, which allows reducing the number of cells of the computational domain. Mayer et al. (1998) and Jacobsen et al. (2012) provide a detailed description of the structure of the relaxation function and of the use of relaxation zones as boundary conditions.

## 3.2 Numerical model set-up

Among the subjects used in the flume experiments, three subjects (i.e. subjects 2,4 and 5) tested by Karvonen et al. (2000), the

subject tested by Jonkman and Penning-Rowsell (2008) and a selection of pairs (*H, U*) of the scale human model used by Xia et al., (2014) were chosen. This selection has been made in order to cover a wide range of flow regimes (both sub-critical and super-critical) and different subject physical characteristics. To generate the mesh around the human body, a free triangulated geometry of a man (STereo Lithography interface format *.stl), downloaded from www.thingiverse.com was used. The heights of the different subjects were adjusted using the 3D scaling functions available in the code for the triangulated geometries.

The mesh domain has a cylindrical shape so that the relaxation zone (with a similar shape and 1.5 m thick) can fully control the generation/absorption of the flood conditions (i.e. water depth and velocity) avoiding possible boundary effects. A mesh sensitivity analysis has been performed with the laminar turbulence model for the numerical simulation of subject 2 (Karvonen et al., 2000) (water depth 0.6 m and velocity 2.0 m/s). Three different mesh sizes around the human surface have been tested:

0.015 m, 0.01 m, 0.005 m. The differences in the estimated drag and lift average coefficients were of the order of a few percent and smaller than the standard deviation of the instantaneous values computed during the simulation. Thus, the 0.015 m mesh has been preferred for its shorter computational time. The total number of cells is around $4.5 \cdot 10^5$. The snappyHexMesh tool allows refining the mesh close to the human body (cell size is set to 0.015 m), while in the whole mesh domain the maximum size is 0.25 m. The refinement close to the human body can be observed in Fig. 2, where the 3-dimensional view of the mesh in a longitudinal cross section is shown for the whole body (panel a), the legs (panel b) and the feet (panel c). The 3D geometry describes a naked body since clothes are difficult to be represented as soft and flexible, thus rigid clothes could affect the estimation of the forces. The time step is set to be automatically adjusted during the simulation according to the maximum Courant number set to 0.7. The order of magnitude was around $10^{-3}$-$10^{-4}$ s to ensure stability. With a time step equal to $2 \cdot 10^{-4}$ s and $4.5 \cdot 10^5$ cells one second of simulation takes 30 minutes without running in parallel (i.e. one core).

The average water elevation H and flow velocity U are initialized in the domain according to the experimental conditions and these values are fixed at the inlet and outlet boundary to a constant value during all the simulation.

The wall function used is the standard nutWallFunction available in OpenFOAM®. The pressure and the velocity fields, needed for the drag and shear forces evaluation, are directly calculated through the continuity and momentum equations (RANS equations) implemented in the model for steady, incompressible and immiscible fluids (Morgan, 2013). The reference 'undisturbed' velocity (U) and an area of reference $A_{ref}$ are set to calculate the instantaneous drag and lift coefficients considering the force acting on the human body in the flow direction, D (Eq. 18), and in the vertical direction, Li (Eq. 18), respectively. Drag force is positive when oriented with the flow and lift force is positive when upward directed. The reference area $A_{ref}$ is the total frontal area of the prism approximating the body normally projected to the flow, equal to $l \cdot H_P$

$$C_D = \frac{D}{0.5 \cdot \rho \cdot U^2 \cdot A_{ref}} \tag{19}$$

$$C_l = \frac{Li}{0.5 \cdot \rho \cdot U^2 \cdot A_{ref}} \tag{20}$$

The total frontal area is selected instead of the wet area because the actual wet area is not simply equal to $l \cdot H$. The determination of the actual wet area would require a dedicated analysis of the flow profile for different flow regimes. Moreover, the total frontal area $A_{ref}$ allows better comparing the pushing efficiency for different submergence levels. However, the selection of the reference area for the hydrodynamic forces is arbitrary and the use of the wetted area is optional. Drag and lift coefficients in the form of Eqs. 19, 20 are derived from dimensional analysis and the reference area is a scale factor with dimensions of (length)$^2$. Thus, wetted area and full frontal area are commonly used in engineering practice (Fox and McDonald, 1978; Hoerner, 1965; Bertin and Smith, 1979).

D and Li include both pressure and viscous forces acting in the flow and vertical direction respectively, although the contribution of the viscous forces is negligible with respect to pressure forces (they differ of six-seven orders of magnitude).

To obtain the force coefficients the time average is calculated once the coefficients have reached the steady state, which is confirmed by the absence of a linear trend.

### 3.3 Tests programme

Three experimental datasets on the instability of people are considered (Karvonen et al., 2000; Jonkman et al., 2008; Xia et al., 2014) because they cover a wide range of flow regimes (i.e. Froude numbers) and include different physical characteristics and a human body model. All simulations account for a frontal impact of the water flow on the human body. Only one flow orientation is considered because most of the experimental studies neglects the effect of the angle of flow incidence. The investigation of a walking condition is considered out of the scope of the manuscript since it would bear different boundary conditions, mesh and working assumptions. The experimental pairs (H, U) recognized as critical in the laboratory tests and used for the numerical simulations are summarized in Table 2 for the different datasets. The experimental data for the human model (Xia et al., 2014) have been scaled to actual size through Froude similarity using the scale ratio λ=5.54. The total number of numerical simulations is 33.

## 4 Results

### 4.1 Forces and force coefficients

The numerical results are analyzed in terms of flow characteristics and hydrodynamic forces. For super-critical flows, a significant splashing area is detected in correspondence of the impact zone (Fig. 3, panels a, c). Figure 3 depicts the simulated flow around the subject tested by Jonkman and Penning-Rowsell (2008) for the pair H=0.35 m, U=2.40 m/s. For this flow condition the free surface elevation decreases downstream after passing the ankles where the flow accelerates, then there is a sudden energy dissipation (behind the ankles, panel b) and the free surface is restored (panel c). The rough aspect of the free surface in Fig. 3 panel a, corresponds to areas with strong mixing between air and water, which has been experimentally observed by Jonkman and Penning-Rowsell (2008).

Panel c in Fig. 3 also shows the distribution of pressures on feet and legs of the subject. Red areas correspond to high pressures located on the inner side of the feet and above the ankles where the flow decelerates. In the external side of the feet depicted in light blue instead, the flow accelerates with a consequent decrease in pressure.

For sub-critical flow condition, the flow is disturbed upstream of the human body, where a slight deceleration occurs. Vortices occur immediately downstream of the obstacle.

Drag and lift forces are integrated over the human geometry during the simulations and the force coefficients are calculated using the frontal reference areas $A_{ref}$ in Table 3, which are evaluated graphically.

Figure 4 shows the drag coefficient and lift coefficients versus Froude number on the right hand side of the figure in the top and bottom panels respectively. Drag coefficient ranges from 0.1 for high Froude numbers, up to approximately 1 for low Froude numbers.

Drag coefficients decrease exponentially with increasing Froude number, i.e. with decreasing submergence. The drag coefficients of all the human subjects (Karvonen et al., 2000; Jonkman and Penning-Rowsell, 2008) are very similar for the same simulated flow regimes. Drag coefficients for the human scale model (Xia et al., 2014) in the range of Froude number 0.4-1.5 appear lower than the coefficients evaluated for human subjects. In fact, the human model is 'weaker' than the real

human subjects in facing the water flow, as demonstrated by the comparison between dimensional thresholds of instability for the model and real humans (Xia et al., 2014). For Froude numbers above 1.5 the drag coefficient for the human model remains almost constant.

Lift coefficients (left and right bottom panels in Fig. 4) range from -0.49 up to 0.06. Except for subject 4, which is represented with a diamond symbol, the lift coefficients are negative. This means that the vertical force contributes to stability because is

directed downward. The two positive values for subject 4 (Karvonen et al., 2000) are due to the relative submergence of the subject $H/H_P$, which is higher than 0.6 (see Fig. 4, bottom left panel). For this level of submergence the water reaches the lower part of the body trunk and thus can exert its action pushing it upward. The negative lift coefficients are the result of the downward directed force acting on the upper boundary of the feet, which is shown in terms of pressures in Fig. 3 (panel c). This occurs because the subject's feet are placed directly on the bottom as a consequence of the assumption of the rigid body.

In actual conditions, when a human subject is allowed to move, the pressure distribution and vertical forces would change significantly because also the sole would experience the hydrodynamic forces.

The left hand side panels of Fig. 4 depict drag and lift coefficients versus the relative submergence $H/H_P$. Drag coefficient increases quadratically as the relative submergence increases since a larger portion of body surface is affected by the water flow, thus increasing the lever arm of the soliciting moment. Moreover, the lift coefficient linearly decreases with increasing

relative submergence.

The human body has a complex shape and its hydrodynamic interaction is affected not only by the flow but also by the portions of the body involved. For the analysed range of water depths, three parts can be distinguished: feet, legs and trunk. Since the lift force is the integral of pressures on the surface, its value is affected by submergence. In fact, for low water depths, legs only contribute to drag force and feet are subject to a vertical force downward directed, given the assumption of adherence

between bottom and feet. Once the pelvis is wetted, and this may occur for undisturbed water depth lower than body trunk due to backwater effects, an upward directed action is added to the downward directed feet action (conventionally negative). With the increase of submergence, the upward component increases until the global vertical force becomes fully positive and this explains the lift force behaviour (Fig. 5).

Figure 5 depicts the lift and drag forces versus Froude number for all the simulated subjects (top and bottom panels

respectively). For human subjects, which have been tested in the range of Froude numbers 0.2-2, drag force increases for 0.2<Fr<1, reaching a peak for Fr~1, then it decreases. The values of drag force for human subjects range from 100 N up to 350 N. Subject 2 tested by Karvonen et al., (2000), which is the tallest and heaviest subject of the dataset, is able to face the highest forces with respect to the other subjects. Subject 4 and 5 are weaker according to the diagram, subject 4 is a woman and subject 5 is a 60 years old man. The estimated forces for the human model (Xia et al., 2014) have been scaled according

to Froude similarity, using the scale ratio 5.54[3]. This allows comparing the dimensional forces of the human model with the forces acting on the human subjects. The behaviour of the human model, whose drag force values are represented in Fig. 5 (bottom panel) with right-oriented triangles, appear different from the human subjects. In fact, drag force values increase linearly with Froude number without reaching a peak for Froude around 1. The peak of drag force observed for human subjects

is the result of a balance between drag-induced moment and immersed weight and ability of actively react to the action of the water flow. Moreover, for Fr=1, where the peak of drag force occurs, the lift force reaches its maximum absolute value (Fig. 5, top panel). Therefore, since the stabilizing effect of the vertical force increases the effective weight, the change of position of real human subjects, with a consequent change of lever arm d (Eq. 13), increases the resisting moment. Thus, a larger drag force can be faced. This is not possible for the human model, since it behaves passively in the water flow without adjusting its

posture.

For Froude number between 0.5 and 1 there is a minimum of the lift force (Fig. 5, top panel), which reaches about -90 N. For low Froude number and relative submergence equal to 0.62 (see Fig. 4, bottom left panel), subject 5 (Karvonen et al., 2000) experiences a positive vertical force since a portion of the lower body trunk is immersed in water. These values correspond in fact, to low values of drag force in Fig. 5 (bottom panel). With the rigid body assumption, for high Froude numbers the human

model is protected by an increasing absolute value of the vertical force, which allows resisting to increasing drag forces. For both drag and lift forces there is a compensation of the opposite effects of submergence and Froude number (i.e. velocity), in fact when submergence increases, force coefficients and Froude number increase (in absolute value) and decrease respectively. Unfortunately, only recently human subjects have been tested for highly super-critical flows (Martinez-Gomariz et al., 2016) so a direct comparison between human subjects and human model was not possible for those regimes. While for sub-critical

flow regimes, it is clear that the ability of human subject to actively resist to the flow adjusting its position is an advantage in terms of safety with respect to the human model. However, the passive behaviour of the human model can be seen as representative of the weakest class of people like elderly or sick as suggested by Xia et al. (2014).

## 4.2 Motion mechanisms

Since literature distinguishes two motion mechanisms, namely sliding and toppling (see Sect. 2), the identification of these

mechanisms is further investigated in this section. The normalized moment is defined as the ratio of drag induced moment and resisting moment, where the effective weight of the subject is calculated subtracting (adding) the vertical force to the weight.

$$Norm\ moment = \frac{D \cdot H}{2(W - B - Li) \cdot d} \tag{21}$$

The normalized moment is represented against Froude number in Fig. 6. In the diagram, there are two regions identified by the calculated normalized moment. As Froude number increases, the submergence decreases in the diagram. In the left side of

the diagram the normalized moment decreases with Froude number until approximately Fr=1.5. Then, for Froude number higher than 1.5 the normalized moment increases slowly. The region with Fr<1.5 is interpreted as the toppling instability area,

while for Fr>=1.5 sliding instability occurs. The separation of the two regions, with low values of normalized moment around Fr=1.5 hints that a mix of the two instability mechanism might occur while approaching Fr=1.5 (Jonkman and Penning-Rowsell, 2008). In fact, for low Froude numbers (i.e. high relative submergence) a full toppling instability is expected; vice versa, for high Froude numbers a full sliding instability takes place.

The identification of the two motion mechanisms helps in defining the dimensionless groups $C_t$ and $C_s$ defined in Sect. 2.2 (Eqs. 10, 16), which are used for the comparison between experiments and numerical results.

### 4.3 Comparison with experimental data

The numerical results obtained from the simulations are compared to the experimental datasets (Karvonen et al., 2000; Jonkman and Penning-Rowsell, 2008; Xia et al., 2014) using the analytical relation between the mobility parameter $\theta_P$, Froude

number and the group accounting for the force coefficients (Sect. 2.2). The groups accounting for the combination of the force coefficients is $C_s$ or $C_t$ for sliding or toppling instability respectively (Eqs. 10, 16). Since the two motion mechanisms have been identified in Fig. 6, $C_s$ is calculated for Froude number equal or larger than 1.5 and $C_t$ for Froude number lower than 1.5. Friction coefficient is assumed constant and equal to 0.3, which is in the range used in literature (Milanesi et al., 2015).

The mobility parameter $\theta_P$ is calculated from the experimental water depth H and Froude number is calculated from the

experimental pairs H, U. The length of the foot assumed for the different subjects is shown in Table 3.

Fig. 7 shows the scatter plot of experimental and numerical results. On the horizontal and vertical axis there are the mobility parameter $\theta_P$ and the product of the square of Fr and $C_s$ or $C_t$ according to the type of motion mechanism.

The determination coefficient is 0.76 and the RMSE is 0.63. The comparison is overall satisfactory given the different data sources, however there are some points which are below the 1:1 curve. Thus, the datasets related to human subjects are

separately analysed since they have shown a different behaviour in terms of hydrodynamic forces. Different symbols represents human subjects (circles) and human scale model (triangles). As expected, the numerical results of the human model compare less well with the mobility parameter and are in general below the 1:1 curve. This is due to the lower estimated drag coefficient/force for human model. Moreover, since the mobility parameter accounts for the full length of the foot d, which is relevant to calculate the resisting moment, its definition may not be appropriate for a human model, which is not able to adjust

its position in order to take advantage of the full length of the foot to react to the instability. If the dataset on the human model is removed, the determination coefficient $R^2$ is 0.82 and the RMSE is 0.28, thus the comparison between numerical model and experiments improves.

A sensitivity analysis to $d$ and friction coefficient $\mu$ has been carried out to understand how a change in these parameters affects the goodness of fit between numerical results and experiments. The parameters $d$ and $\mu$ play a role in the calculation of $C_t$ and

$C_s$ respectively (Eqs. 16 and 10). A variation of $\pm$ 10% and $\pm$ 30% has been applied to both $d$ and $\mu$ one at time and the change in the determination coefficient $R^2$ and RMSE of the fit has been calculated. The results are summarized in Table 4. The results of the sensitivity analysis are overall satisfactory since the determination coefficient $R^2$ does not decrease significantly when modifying the parameters $d$ and $\mu$ both considering only experiments on human subjects both considering all the datasets. In

fact, the determination coefficient $R^2$ does not decrease under 0.7. The RMSE tends to increase for larger variations of the parameters especially considering all datasets. A more accurate comparison between numerical results and experiments would be possible if friction and length of the foot were measured during experiments, which is strongly encouraged in future research.

## 5 Discussion

Flood hazard and flood risk maps as required by the European Flood Directive 60/2007/EC (European Commission, 2007), should identify the areas which can be affected by floods for different probability scenarios and their potential adverse

consequences on the environment, structures and people. Nevertheless, despite the increased capability of hydrologic-hydraulic modelling and damage assessment models, the direct consequences of flood parameters (e.g. water depth and velocity) on human health are often overlooked in hazard and risk maps. This is also due to sparse research on the subject and to the difficulties in identifying precise relationships between flood characteristics and people instability. Usually different hazard zones are classified according to the product number $H \cdot U$ (Cox et al. 2010). The curves so defined attempt to interpret the

large scatter observed in dimensional pairs of water depth $H$ and velocity $U$ in which instability occurred in flume experiments, but are not capable of discerning stability conditions among different individuals.

For this reason in the paper, a dimensionless instability criterion for people under water flow has been proposed. The mobility parameter $\theta_P$ is a function of the physical characteristics of the human subject (i.e. height $H_P$ and length of the foot d). It also shows a strong dependence with Froude number and accounts for the two recognized instability mechanisms, which are sliding

and toppling. The evaluation of the mobility parameter for a selection of experiments available in literature (Foster and Cox 1973, Karvonen et al. 2000, Jonkman and Penning-Rowsell, 2008 and Xia et al. 2014) identifies a unique threshold for people instability $\theta_{Pcr}$ capable of reducing the scatter of dimensional critical combinations of water depth and velocity. Since $\theta_P$ is dimensionless, it allows comparing the instability conditions for vehicles (Arrighi et al., 2015) and people in the same dimensionless diagram (Fig. 8).

The critical dimensionless threshold curves drawn in Fig. 8 for vehicles (black continuous line) and people (black dashed line) intersect for Froude number approximately equal to 0.6. Thus, four different portions (i.e. hazard zones) can be observed in the diagram. Above both the curves both pedestrians and vehicles can be classified as stable for a given flow regime (i.e. Froude number). Below the curves instead, both people and vehicles are in unstable, and consequently dangerous, conditions. For practical applications and risk mapping a safety factor could be applied to shift down the threshold curves and account for

hydraulic model uncertainties and experimental variance. For Froude number between 0.1 and 0.6 moving in floodwaters by car is safer than moving on foot since the $\theta_{Pcr}$ threshold curve lies above the $\theta_{Vcr}$ curve. For Froude number above 0.6 the $\theta_{Vcr}$ curve lies above the $\theta_{Pcr}$ curve for people, thus, for these flow regimes moving on foot is better than use a car. In simple words,

wading a creek is safer on foot, wading a shallow river is safer by car. In fact, the two curves $\theta_{Vcr}$ and $\theta_{Pcr}$ show the different dominant modality of instability for vehicles and pedestrians, which depend on the different geometric configuration and mass distribution. For low Froude number the dominant instability mechanism is toppling, to which pedestrians are more vulnerable than vehicles. For high Froude numbers sliding instability prevails, which, in the case of pedestrians is counterbalanced by a

lower lift effect, in the case of vehicles instead contributes to a lower adherence. Since higher Froude numbers in the diagram correspond to lower water depths, this result may be not intuitive for a person facing a flood flow. In fact, hazard in low water depths is usually underestimated. In fact, for lower water depths, which can be felt as less threatening, a person can be induced to move by car, which is perceived as a safe shelter. That is why education can be of crucial role.

A more popular version of this diagram may help supporting people education because it clarifies the instability mechanisms

of vehicle and people, which are recognized as responsible of most of the casualties. Moreover, the critical thresholds here proposed can be easily coupled with existing flood maps adding further information on hazard levels to be adopted for mitigation strategies and emergency activities.

The 3D numerical model, although very simplified since the human body is modelled as rigid, is the first example of numerical investigation on the instability conditions of people in under water flow. It demonstrates the importance of people ability of

counteracting the hydrodynamic forces, through the adjustment of the posture. In fact, the forces evaluated for the instability conditions of the human scale model (Xia et al., 2014) appear lower than those for human subjects. As suggested by Xia et al. (2014) these conservative conditions can be adopted to account for particularly weak categories of people like elderly or sick. Moreover, the numerically evaluated forces show that subjects with larger weight and height are able to resist to higher solicitations confirming the observed experimental variability between the subjects (Cox et al., 2010, Russo et al., 2013).

Further experiments on human subjects should investigate the instability conditions in super-critical flow regimes, which have been currently addressed only by Jonkman and Penning Rowsell (2008) and recently by Martinez-Gomariz et al. (2016). The evaluated force coefficients, which are a dimensionless measure of the forces, are strongly similar for the different human subjects and can be adopted in conceptual models, which usually account for standard values for cylinders.

Further studies should better investigate the role of friction coefficient for the occurrence of instability (Martinez-Gomariz et

al., 2016), which might be crucial especially for supercritical flow regimes. Moreover, the effect of different physical (i.e. body type, size and build) and psychological human characteristics on the hydrodynamic solicitations should be better understood as well as the role of relative motion, posture, and clothing. Then, more detailed laboratory experiments and numerical model, with turbulence measured and accounted for, could investigate important environmental aspects, such as local turbulence effects (Chanson and Brown, 2015). A more reliable estimation of the hydrodynamic forces on the human body could be

achieved removing the strong assumption of rigid body and feet-bottom adherence hypothesis. This would suggest the use of a fully coupled CFD-CSD model capable of accounting for the different hydrodynamic response to changes in posture.

**6 Conclusions**

People safety is the primary objective for flood risk managers in the definition of non-structural risk mitigation measures. Numerous studies demonstrated that most of the casualties for drowning during a flood occurs because of unwise high-risk behaviors like driving and walking in floodwaters. Current hazard zoning rely on the product number H·U, which helps in explaining the large scatter of experimental pairs of water depth and velocity found in the last decades. However, the H·U criterion is empirical and neglects subject characteristics, whose variability is not physically accounted for.

This paper provides a new approach for hazard assessment of people in floodwaters. The dimensionless mobility parameter here introduced, calculated for selected existing experimental datasets, is capable of identifying a unique critical threshold of instability $\theta_{Pcr}$ regardless of the type of motion mechanism (i.e. sliding and toppling), which is a function of relative submergence and Froude number. The scatter of dimensional experimental data is reduced because the mobility parameter $\theta_P$ accounts for both flood (H, U) and subject characteristics (height $H_P$ and length of the foot d). The diagram of Fig. 8 allows risk management specialists assessing pedestrians' instability through the comparison between $\theta_P$ and critical threshold $\theta_{Pcr}$ thus distinguishing different individuals from their height. Thanks to its dimensionless definition, the mobility parameter for people can be compared to the existing mobility parameter for vehicles (Arrighi et al. 2015). Thus, it can support the development of behavioral rules conceived for people education. Moreover, it can also be mapped over existing flood hazard maps showing water depth and velocity, for an average subject used as a reference or with a probabilistic distribution of human characteristics. The sensitivity analysis carried out with respect to geometric and density parameters of the human subjects hints that $\theta_P$ is robust since the length of the food *d,* which is the most sensitive parameter, may vary in a very small range according to allometry observations.

The 3D numerical model presented in this paper, although simplified, demonstrate through the evaluation of the hydrodynamic forces and force coefficients that relative submergence and Froude number are the most relevant dimensionless parameters for people instability. The human body is modelled as rigid and is described by a detailed 3D triangulated geometry. 33 steady flow numerical simulations have been carried out to reproduce three different experimental datasets (Karvonen et al. 2000, Jonkman and Penning-Rowsell, 2008 and Xia et al. 2014) and subject characteristics, covering a wide range of flow regimes (i.e. Froude between 0.2 and 3.5). The numerical results also clarified the different behaviour of human subjects and human scale model. A further study, both numerically and experimentally, should better investigates the role of other aspects, which affect people instability in flood waters, such as local turbulence effects, friction, relative motion, posture, clothing and water density.

**Team list**

The team is composed by the three submitting authors.

**Author contribution**

The research work was designed and carried out by the first author during the PhD activities under the joint supervision of the two co-authors.

**Acknowledgments and Data**

A Ph.D. research fellowships for the first author is provided by the University of Florence, Italy. The original data about people instability can be found in the cited papers, whose authors are acknowledged for providing an experimental substrate to this work. Authors gratefully thank the editor Dr. Hannah Cloke and the anonymous reviewers for their remarks and suggestions, which helped improving the quality of the manuscript.

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

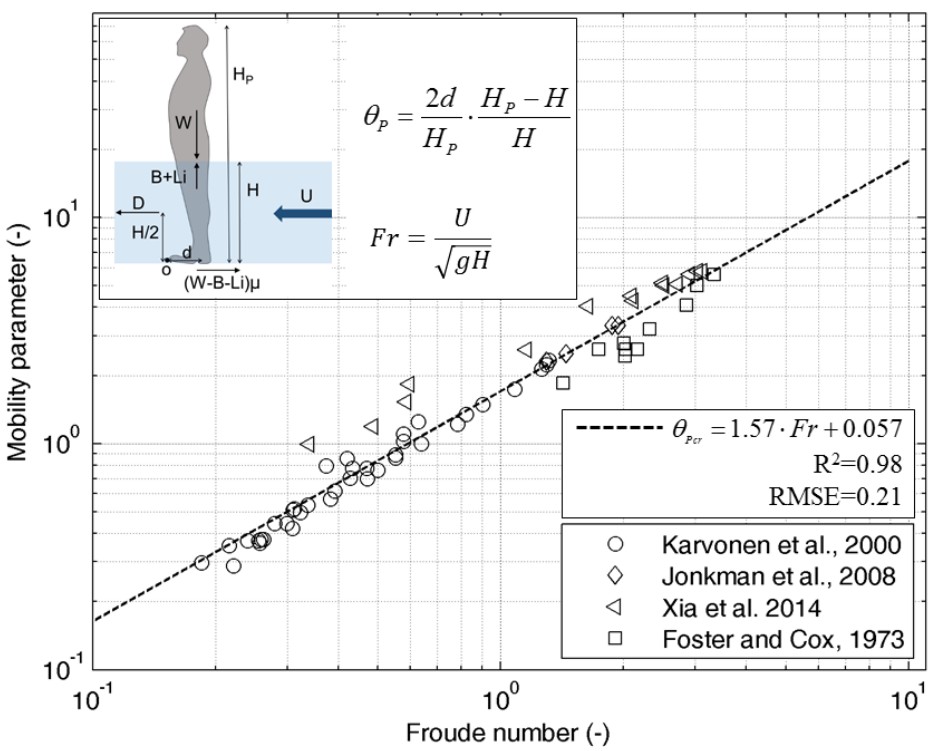

**Figure 1. Instability diagram "Dimensionless mobility parameter $\theta_P$ versus Froude number" for the selected studies (Foster and Cox 1973, Karvonen et al. 2000, Jonkman and Penning-Rowsell 2008 and Xia et al. 2014).**

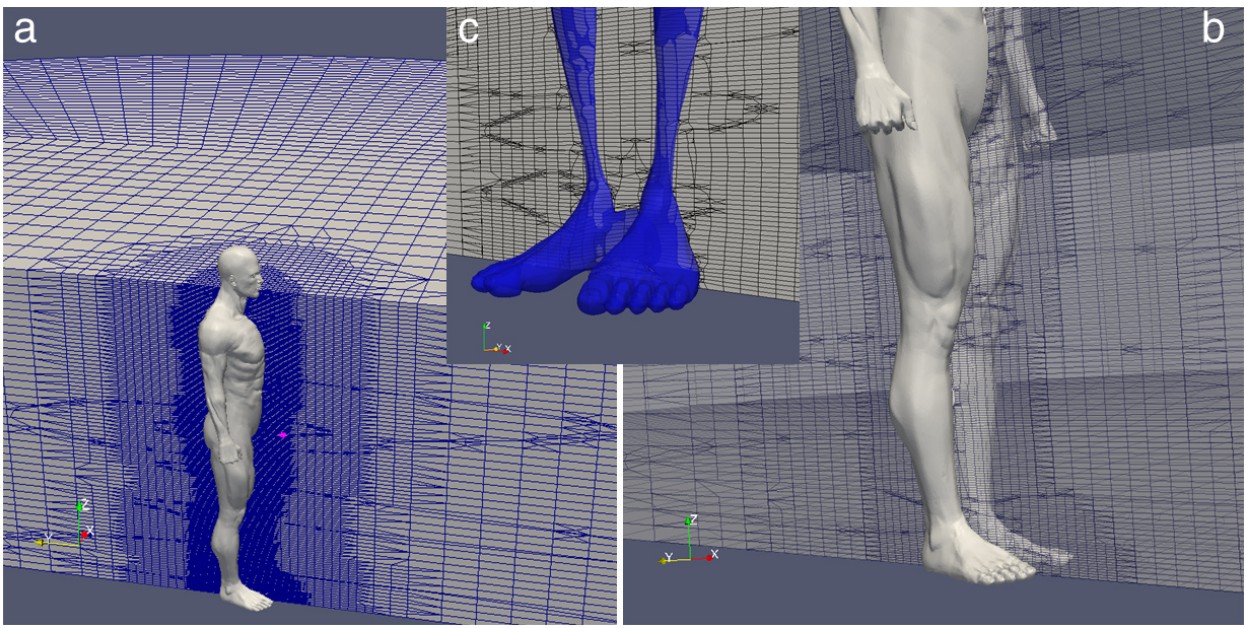

**Figure 2. Computational mesh around the human body shown in a longitudinal cross section for the whole body (a), the legs (b) and a detail of the feet (c).**

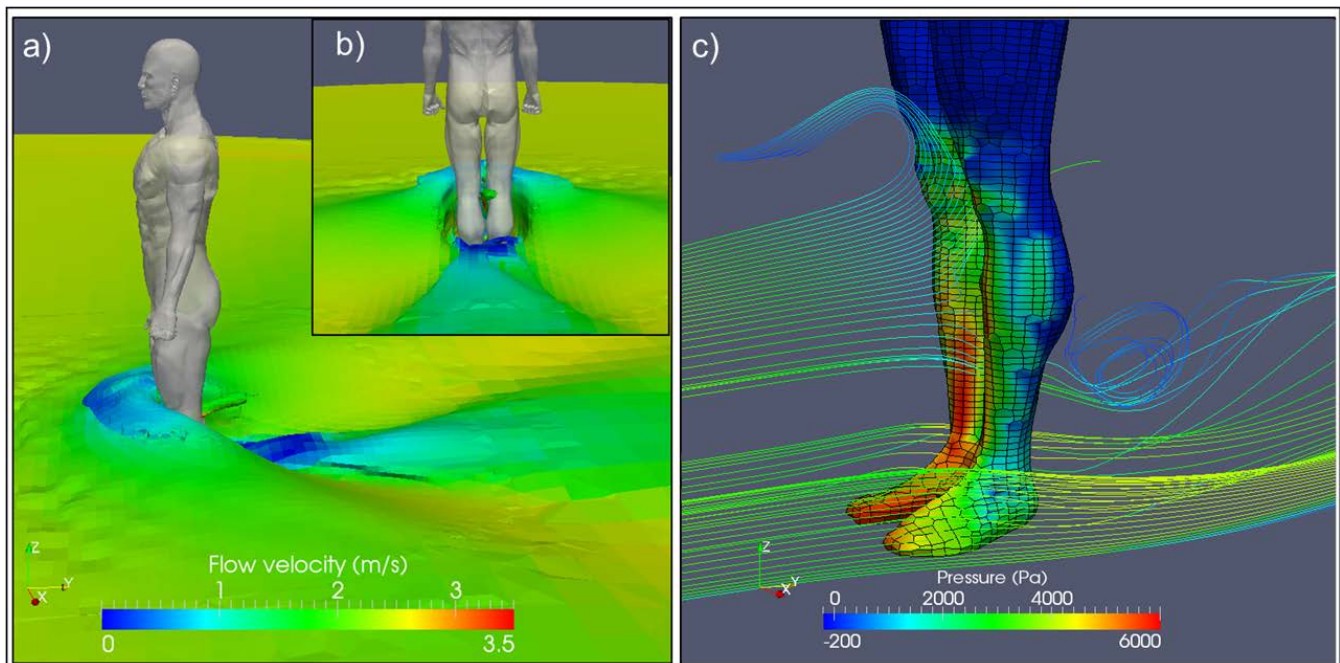

**Figure 3. Splashing effect for super-critical flows shown as flow velocity (a), streamlines (c) and inset view parallel to flow direction upstream-oriented (b), for the subject tested by Jonkman and Penning-Rowsell (2008), H=0.35 m, U=2.40 m/s. Panel (c) also shows the pressure distribution on the feet and the legs of the subject.**

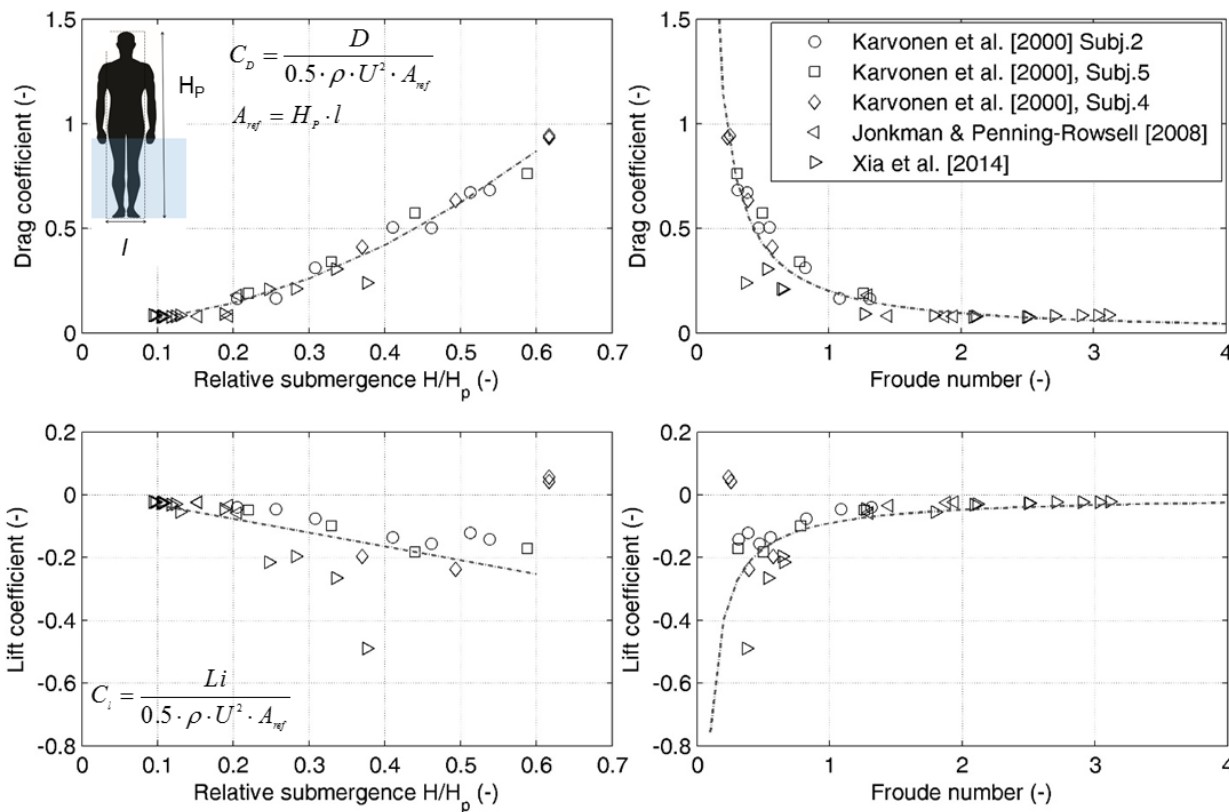

**Figure 4. Estimated drag and lift coefficients versus Froude number (top left and bottom left panels respectively) and versus the relative submergence (top right and bottom right panels respectively) for the four human subjects (Karvonen et al., 2000; Jonkman and Penning-Rowsell, 2008) and the human scale model (Xia et al., 2014). The inset in the first subplot shows the reference area used for the force coefficients calculation.**

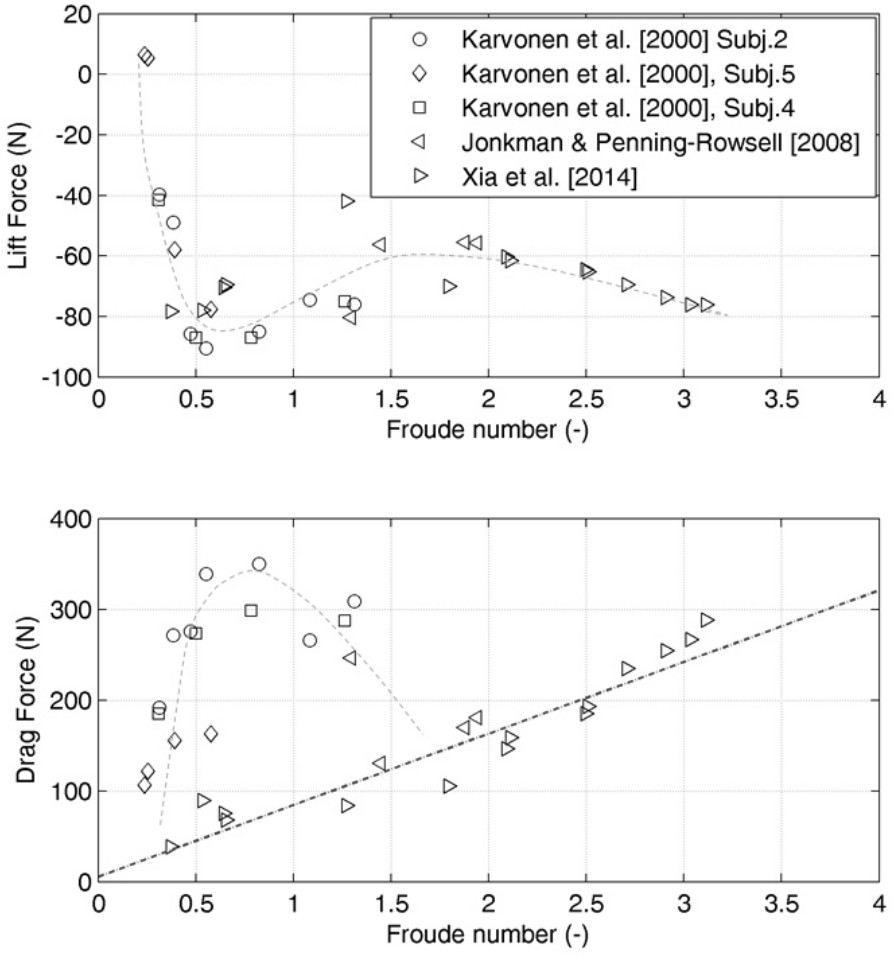

**Figure 5. Estimated lift (top panel) and drag (bottom panel) forces versus the relative submergence for the four human subjects (Karvonen et al., 2000; Jonkman and Penning-Rowsell, 2008) and the human scale model (Xia et al., 2014).**

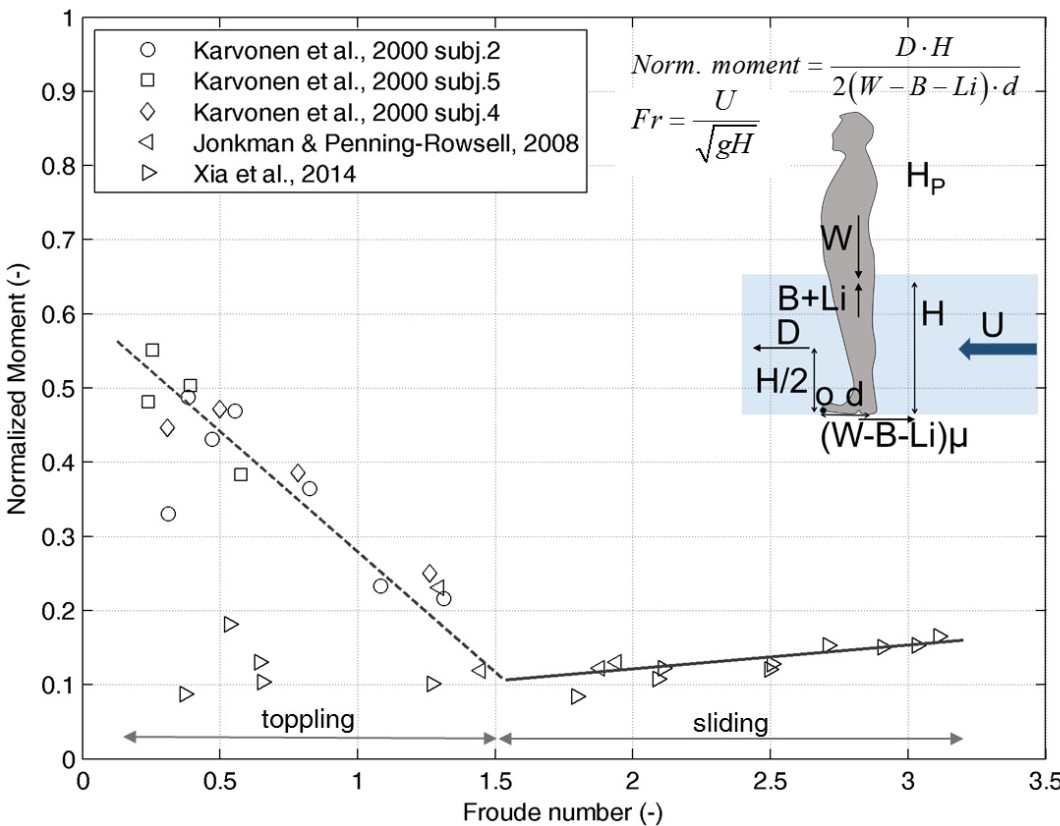

**Figure 6. Normalized moment against Froude number for the four human subjects (Karvonen et al., 2000; Jonkman and Penning-Rowsell, 2008) and the human model (Xia et al., 2014).**

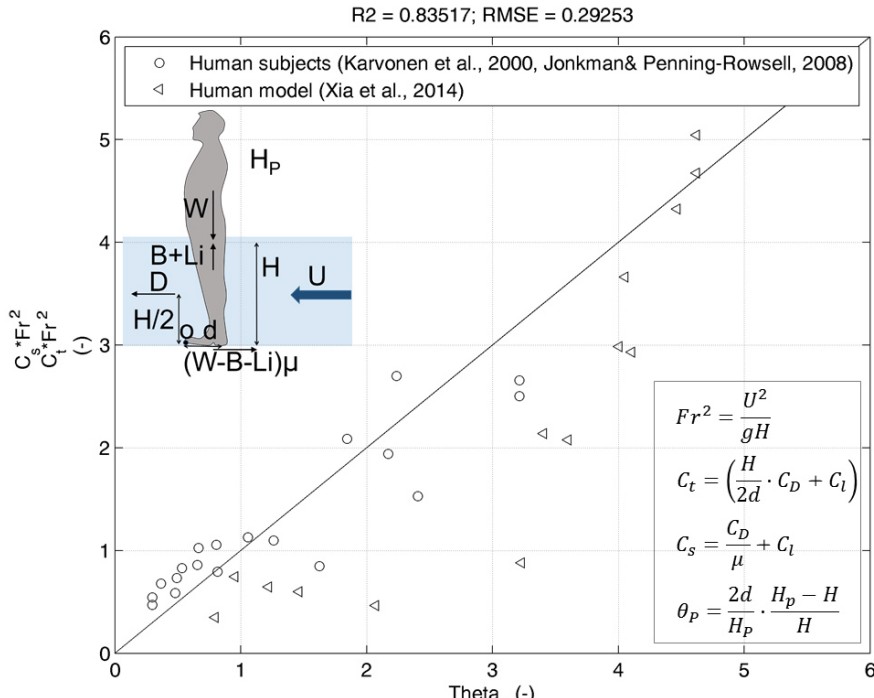

**Figure 7. Comparison between experiments in terms of mobility parameter $\theta_P$ and numerical results, with a distinction between the human subjects (circles) and human model (triangles). In the top of the Figure the determination coefficient $R^2$ and RMSE for the data on human scale model Xia et al. (2014) removed.**

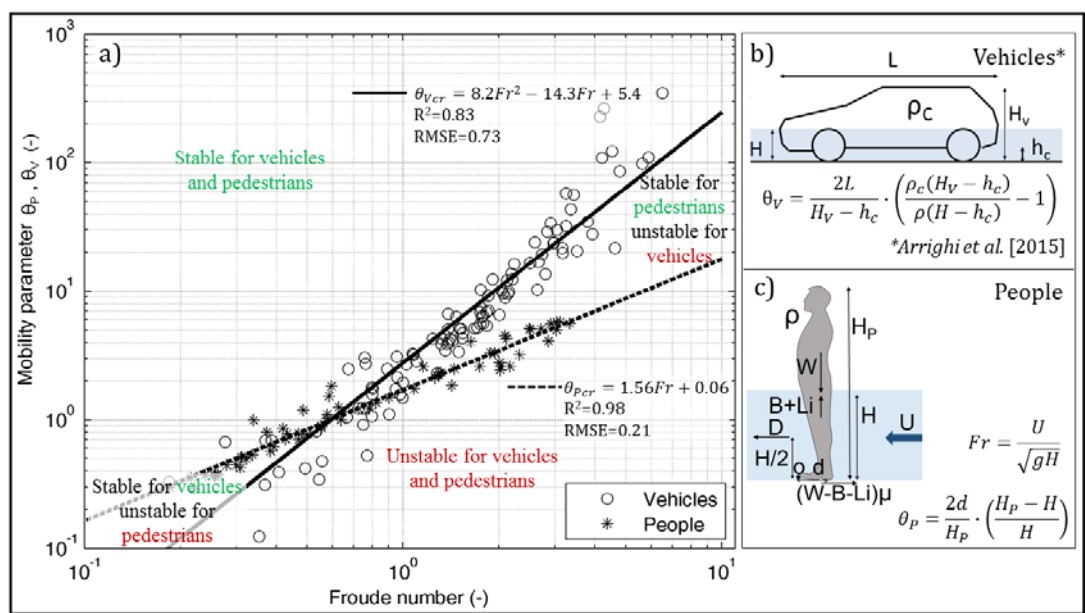

**Figure 8. Comparison between dimensionless mobility parameter for people instability in flood flows θP and dimensionless mobility parameter for incipient motion of flooded vehicles (Arrighi et al. 2015) (a), definitions of the parameters and geometric sketches for vehicles (b) and people (c). The black continuous and dashed lines represent the critical dimensionless incipient motion curve for flooded vehicles and people respectively.**

**Table 1. Sensitivity of the general θ_P (Eq. 12) with respect to geometry and density parameters**

| | Parameter for the sensitivity $X_j$ | | |
|---|---|---|---|
| | $X_j=d$ *length of the foot* | $X_j=H_p$ *height of the subject* | $X_j=\rho_P$ *human body density* |
| Sensitivity function $\dfrac{\partial \theta_P}{\partial X_j}$ | $\dfrac{2\rho_p}{\rho H} - \dfrac{2}{H_P}$ | $\dfrac{2d}{H_P^{\,2}}$ | $\dfrac{2d}{\rho H}$ |

**Table 2. Simulated pairs of water depth H and flow velocity U for human subjects and scale model (Karvonen et al., 2000; Jonkman and Penning-Rowsell, 2008; Xia et al., 2014)**

| Water depth H (m) | Flow velocity U (m/s) | Froude number Fr (-) |
|---|---|---|
| Karvonen et al. (2000), subject 2 | | |
| 0.40 | 2.60 | 1.31 |
| 0.50 | 2.40 | 1.08 |
| 0.60 | 2.00 | 0.82 |
| 0.80 | 1.55 | 0.55 |
| 0.90 | 1.40 | 0.47 |
| 1.00 | 1.20 | 0.38 |
| 1.05 | 1.00 | 0.31 |
| Karvonen et al. (2000), subject 4 | | |
| 0.6 | 1.4 | 0.58 |
| 0.8 | 1.1 | 0.39 |
| 1 | 0.75 | 0.24 |
| 1 | 0.8 | 0.26 |
| Karvonen et al. (2000), subject 5 | | |
| 0.40 | 2.50 | 1.26 |
| 0.60 | 1.90 | 0.78 |
| 0.80 | 1.40 | 0.50 |
| 1.07 | 1.00 | 0.31 |
| Jonkman and Penning-Rowsell (2008), stuntman | | |
| 0.26 | 3.00 | 1.88 |
| 0.26 | 3.10 | 1.94 |
| 0.33 | 2.60 | 1.45 |
| 0.35 | 2.40 | 1.30 |
| Xia et al. (2014), human model (Froude scaled H,U) | | |
| 0.64 | 0.85 | 0.34 |
| 0.57 | 1.15 | 0.49 |
| 0.48 | 1.27 | 0.59 |
| 0.42 | 1.21 | 0.60 |
| 0.32 | 2.05 | 1.16 |
| 0.22 | 2.40 | 1.63 |
| 0.21 | 3.03 | 2.11 |
| 0.20 | 2.93 | 2.09 |
| 0.18 | 3.35 | 2.51 |
| 0.18 | 3.30 | 2.50 |
| 0.18 | 3.60 | 2.71 |
| 0.17 | 3.70 | 2.91 |
| 0.16 | 3.80 | 3.03 |
| 0.16 | 3.90 | 3.11 |

**Table 3. Reference areas for force coefficients calculation and physical characteristics (height Hp, weght W and length of the foot d) of human subjects and human scale model.**

| Subject | N°2 Karvonen et al. (2000) | N°4 Karvonen et al. (2000) | N°5 Karvonen et al. (2000) | Jonkman and Penning-Rowsell, (2008) | Model scale 5.54, Xia et al. (2014) |
|---|---|---|---|---|---|
| $A_{ref}$ (m$^2$) | 0.49 | 0.42 | 0.46 | 0.43 | 0.014 |
| $H_p$ (m) | 1.95 | 1.62 | 1.82 | 1.7 | 0.31 |
| W (kg) | 100 | 57 | 94 | 68.2 | 0.334 |
| d (m) | 0.30 | 0.25 | 0.28 | 0.26 | 0.048 |

**Table 4. Sensitivity of the goodness of fit to the parameters d and μ.**

| Exp. | Human subjects | All data | Human subjects | All data | Human subjects | All data | Human subjects | All data | Human subjects | All data |
|---|---|---|---|---|---|---|---|---|---|---|
| | Base case $d=0.15H_p$ | | $d$ +10% | | $d$ -10% | | $d$ +30% | | $d$ -30% | |
| $R^2$ | 0.82 | 0.76 | 0.82 | 0.73 | 0.81 | 0.70 | 0.81 | 0.64 | 0.77 | 0.62 |
| RMSE | 0.28 | 0.67 | 0.29 | 0.67 | 0.31 | 0.69 | 0.31 | 0.97 | 0.32 | 1.26 |
| | Base case $\mu=0.3$ | | $\mu$ +10% | | $\mu$ -10% | | $\mu$ +30% | | $\mu$ -30% | |
| $R^2$ | 0.82 | 0.76 | 0.82 | 0.72 | 0.82 | 0.74 | 0.78 | 0.69 | 0.80 | 0.71 |
| RMSE | 0.28 | 0.67 | 0.25 | 0.68 | 0.32 | 0.67 | 0.22 | 0.71 | 0.46 | 0.75 |