# Peer review of "Hydrodynamics of pedestrians' instability in floodwaters"

_Hydrology and Earth System Sciences, 2016_

## Short Comment (SC1) · 5 Jul 2016

First of all I would like to congratulate the authors because of the great work carried out. I think that the uses of 3D simulations can help us a lot to figure out many problems, specifically those related to flood risk management as you well describe within this manuscript. I have had the chance to read the entire document deeply and I have some doubts and suggestions that I would like to share with you.

- A comprehensive and recent work that you passed over is "Experimental study of the stability of pedestrians exposed to urban pluvial flooding" by Martínez-Gomariz et al. (2016). In fact, all these "non-hydraulic" parameters mentioned in your manuscript were analyzed within the cited article and widely described in there. http://link.springer.com/article/10.1007%2Fs11069-016-2242-z

[Figure]

- In my opinion the length of the foot as a lever arm (d) is an assumption that depends too much of the flow orientation. According to the derivation of the dimensionless mobility parameter for toppling instability, the length of the foot is even conservative when a right-to-left flow is considered. Therefore, a much lower "d" value would affect for left-to-right flow. In fact, your parameter derivation is based on a right-to-left while the simulations were carried out by considering the opposite orientation. Maybe some explanations regarding this matter should be offered or any sensitivity analysis of flow orientation effect should be conducted.

- In my experience, results of forces obtained through 3D simulations are quite sensitive not just to the size cell but as well to the turbulence model choice. In that sense, either a non-adoption turbulence model or a "laminar" consideration sounds a bit weird to me. In my opinion, at least one of the simulations should be undertaken with any turbulence model and compare the results against the "laminar" results.

- Line 31pg12: Unfortunately, no human subjects have been tested so far for highly super-critical flows so a direct comparison between human subjects and human models is not possible for those regimens → I refer in that sense as well to Martínez-Gomariz et al. (2016) work.

- Line 20 and line 24 of pg 15. You are referring to both needs: more experiments and the role of friction coefficient understanding. In that sense I refer to Martínez-Gomariz et al. (2016) work again. Actually, after some investigation in this field I could figure out that the friction coefficient between the shoe sole and the terrain depends not just on the materials of both but as well on the position of the human body during the walking process. There are, in fact, many studies which analyze people's falls and a good reference could be: Haslam, R., Stubbs, D. (2006). Understanding and preventing falls. CRC. London, New York. For sure these works are not related to people walking through water flows but those fit conceptually perfectly.

---

## Short Comment (SC2) · 14 Jul 2016

Thank you for reading and posting comments to my manuscript, your suggestions and doubts are very appreciated and will contribute to improving the quality of the manuscript. I would also like to thank you for informing me about the recent publication of your manuscript, which, as you pointed out, investigates the pedestrians' instability for supercritical flow regimes. Your work fills a gap in the state of art, which helps the better understanding of the physical phenomenon. Therefore, it deserves to be quoted in my manuscript, whose state of art was preceeding the publication of your work. However, I would like to highlight that the experiments on people's instability in floodwaters in case of walking or standing subjects are quite different in terms of mechanical assumptions and hydrodynamic buondary conditions. I will try to reply point by point to your comments.

- Lever arm: The length of the foot (d) was assumed as lever arm of the resisting moment in order not to underestimate the ability of a subject of adjusting its posture when it feels a stress. Obviously, selecting a 'rigid' lever arm for an object, which is flexible and has many degrees of freedom is an hard task. With reference to the inset of Fig. 1, if the flow comes from right to left we expect (as argued by the previous experimental tests) that the subject will move towards the incoming flow with a consequent displacement of its center of mass. The magnitude of this displacement is extremely variable and the full length of the foot was preferred to other possible lever arms in order to introduce a sort of simplifying simmetry for the two opposite flow directions. I plan to modify the inset picture of Fig.1 to better explain this assumption and also to add a more precise discussion in this topic. About the numerical simulations, some tests have been carried out also for a flow coming from left to right (with reference to the inset of Fig.1), confirming the negligible influence of flow direction in the estimated hydrodynamic forces.

- Numerical model: The 'laminar' settings of the code do not simulate a laminar flow, which would be of course not physically meaningful, they simply avoid to model turbulence effects with a predefined closure model. With this assumpton the drag coefficients evaluated on a circular cilinder differed from the literature values in the same Reynolds number range of approximately 5-8%. The 'laminar' assumption was thus considered acceptable since the selection of non appropriate coefficients for the closure models would bear similar errors in the force estimation. A further work should consider both experimental and numerical experiments to correctly calibrate and tune the turbulence coefficients.

- Line 31 p12: I will definitely modify this sentence and quote your paper.

- Line 20 and 24 p25. As above, thank you for your experimental activity and results about friction and role of non-hydraulic parameters, the paper will be quoted there. Let me mention again the difference in hydrodynamic boundary conditions between standing and walking subjects. The numerical model and dimensionless parameter
introduced only the case of an upright standing subject and no conclusions can be drawn at the moment on the case of a walking subject, which should be modelled numerically by an extremely complex coupled CFD-CSD model. However I would be glad to use the experimental data you plotted in Fig. 9 (high hazard-medium hazard) of your paper together with the corresponding heights of the subjects in order to improve the final discussion of the manuscript.

---

## Referee Comment (RC1) · Anonymous Referee #1 · 16 Jul 2016

This is a nice paper describing a new hydrodynamic model simulating pedestrians' instability during flooding events. The manuscript includes nice figures and is overall well written, but a double check from a English native speaker would help improve readability. Yet, I have 2 main major concerns that should be address before publication.

The first one is related to the actual usefulness of such a model in risk management. The authors state this point in various parts of the paper, but they do not clarify how such a model can actually be used. I think this clarification is crucial, as its usefulness is one of the selling points of such a model.

The second one is related to the lack of a sensitivity analysis. The current modelling exercise does not allow a proper evaluation of the fact that good results are obtained for the right reasons. The paper describes it only with reference to mesh resolution, but

there are numerous parameters affecting model outcomes and, by playing with them, one can get a plethora of different results. Thus, to evaluate the proposed model a comprehensive sensitivity analysis is crucial.

---

## Short Comment (SC3) · 22 Jul 2016

Thank you for the interest in our manuscript and for your comments.

An English check will be included, as you suggested, during the final review.

Regarding you comment about the usefulness of the model, I would say that the main contribution of the work is the identification of the most relevant dimensionless actors playing a role in pedestrians instability. The first and most important one is the relative submergence, which accounts for a characteristic of the subject (height) and for a characteristic of the flow (water depth). The second parameter is Froude number, which is an attribute of the flow. Moving from conventional diagrams (water depth versus velocity) to dimensionless diagrams may help in the definition of safety rules for citizens. This doesn't mean that common people should read a dimensional diagram.

As an example, teaching people to recognize a level of submergence with respect to their body (knees, ankles, waist) is easier than refer to absolute water depths, which are more difficult to assess. This point will be better clarified in the manuscript during the review process.

The 3D numerical model introduce in the paper is affected mostly by the assumption of the rigid body as explained in the comparison between the numerical results of human subjects and human model. Since turbulence is not modelled for the absence of adequate calibration/validation phase for the coefficients of the closure model, the parameters involved are water viscosity, gravity and mesh size. Therefore the effect of the mesh size has been evaluated. Again, it is necessary to mention that the purpose of the numerical model is not to evaluate the 'exact' value of the hydrodynamic forces, but to understand their overall contribution over a wide range of flow regimes and submergence levels. If the referee invokes the sensitivity analysis with respect to other parameters such as flow orientation or body build type, this should be left for a comprehensive ad hoc parameter study.

---

## Referee Comment (RC2) · Anonymous Referee #2 · 29 Jul 2016

The authors introduce an interesting topic that has been addressed by several authors before through an experimental point of view. In the paper there is basically a numerical modelling considering a 3D approach, and some hypothesis associated to it.

Some comments of the authors concerning the scatter observed in the experimental studies, must be clarified. Most studies consider different types of individuals to test their problems against urban floods. And it is perfectly logical this scatter, and consider to define hazard criteria as a the lower bound values, not the highest ones. Authors for instance take part of the data of the studied developed by Karvonen, oriented to define hazard criteria, not for the normal people living in the urban area, but for rescue services, people much more trained and strong than aged people or teenagers. So to propose any function including this kind of data, could be biassed not to the normal people. Same comments concerning the message about the tests developed by Foster

and Cox, including babies. So it is logical that limiting conditions are not the same for everybody, babies and rescue services, and the scatter must be observed as it is in the real tests. It would be interesting if authors include all the data concerning the previous test developed by other authors (Abt, Takahashi, etc), and their statement that "... scatter can be reduced—" considering their parameter is fully right or not.

Authors have made their choice considering just one orientation, person parallel to the flow. In flood events most of the problems are associated to the crossing of flooding streets. In those cases some of the distances considered in the analysis would be quite different, making more dangerous this condition and not the considered in the paper.

Concerning the application of the 3D code, I would like to get some more details. Authors indicate that no turbulence model has been considered. Especially for local effects, turbulence closures are needed to reproduce properly what really happens. The reason why the authors say they do not consider a turbulence model is not clear for me. Results must be checked with a turbulence model and compare if results change or not. Authors calculate few seconds. Is it fully stable flow? Some authors indicate problems when turbulence models are introduced and forces are calculated, and the process leads to some instabilities. Data about CPU time is welcome for the tests made

And another point not fully clear to me is the consideration of the lift force. Authors make the distinction between the buoyancy and lift force (ec. 1). But next they consider density equals between water and humans. What if not? More physical explanations concerning the final orientation are these forces would be welcome. And more interpretation of the results considering the huge variation of drag and lift coefficients is welcome too. Shape of the "person" tested is the same, so for instance drag coefficient can change with a ratio close to ten, for different rates of submergence. At the beginning seems reasonable, but when the shoes/ foot are covered, increase of submergence affects only to legs, so shape is almost the same.

The authors present a variation of the Lift force with Froude number, showing a first decrease, then an increase and another decrease. Can they introduce some physical explanation to this point?

Errata: Page 13, line 9, there are two consecutive "this".

---

## Author Comment (AC1) · 1 Aug 2016

On behalf of my co-authors I would like to thank the anonymous referee #2 for his comments and his interest in the manuscript. We absolutely agree with the referee on the first point concerning the scatter of existing experimental data. It is perfectly understandable why experiments carried out on different human subjects (i.e. children or professional stuntmen) differ in the corresponding pairs of water depth and velocity. We also stress these 'dimensional' differences in terms of estimated forces while commenting Fig.5 for instance. In the manuscript we are not arguing that the scatter is not physically meaningful, but, given the existing scatter, we investigate the hypothesis that a limited number of dimensionless parameters can explain most of the existing variability. In our opinion, the mobility parameter and the dimensional analysis are capable of identifying two main actors in the phenomenon of instability of pedestrians in

floodwaters, which are relative submergence and Froude number. The contribution of these parameters is then investigated through a 3D numerical model. The discussion on the numerical results also include the effect of other parameters, which were not investigated in this study, but are left for a future research. Regarding the use of the data by Abt et al. and by Takahashi, in the first case we considered the experimental conditions not fully representative of a 3D flow and limited in the investigated range of flow regimes, in the latter it was not possible to retrieve the heights of the tested subjects to calculate the mobility parameter. In order to better clarify these points we will add some new paragraphs in the final revised version.

It is true that we only considered one flow orientation as a simplifying assumption; also most of the existing experimental studies neglect the effect of the angle of flow incidence. We also considered a rigid body thus we neglect the walking condition. As replied to E. M. Gomariz, the boundary conditions and set up of the 3D numerical model should be completely modified to account for a non-rigid or moving body and this is out of the scope of the manuscript. We will add a further discussion to clarify this point.

The 3D model aims at investigating the role of the mean flow properties on the phenomenon of instability and a turbulence model was not selected for two main reasons: first, a turbulence model needs the calibration and/or validation of some coefficients and adequate experiments were not available for this purpose; second we are approximating with a rigid body the experimental conditions for subjects which were allowed to move freely. This obviously bears an error in the estimation of the forces on the subject, thus we considered our assumption adequate for the intrinsic uncertainties of the simulated problems. Moreover, as stated in the manuscript, the results of the simplified numerical model were acceptable for a circular cylinder used as a benchmark. The stability of the flow was checked and the parameters were extracted once the simulation reaches steady conditions. The time required for simulating 1 s was about 30 minutes with one core and important reductions of the computational time were achieved

through parallelization. We will better clarify the reasons of our hypothesis and the other points in the manuscript.

Assuming the water density equal to the human density is quite common in the conceptual models since the density of floodwater is reasonably higher than freshwater density, due to suspended solid transport. From the physical point of view, this assumption means that in fully submerged conditions the human body lies in static equilibrium. About the lift force orientation we will add a clearer explanation in the text, however the lift force is oriented downward as long as the only horizontal surface involved in the flow is that of the feet, it becomes upward-oriented when the lower part of the body trunk is reached by the flow. The interpretation of the variation of drag and lift coefficients will be clarified highlighting in the diagrams (Fig. 4) how the force coefficients were initially defined. This should resolve the ambiguity of the variation of the forces and force coefficients with the submergence and help in the overall understanding of the manuscript.

---

## Referee Comment (RC3) · Anonymous Referee #3 · 13 Aug 2016

**Manuscript Number**: hess-2016-261

**Title:** Hydrodynamics of pedestrians' instability in floodwaters

**Hydrology and Earth System Sciences**

**General Comment:**

People safety can be compromised when they are exposed to floodwaters that exceed their ability to remain standing, thus leading to a great loss of casualties. Therefore, it is important to study the pedestrian's instability in floodwaters. In order to overcome the scatter of existing experimental data, the authors introduced a dimensionless mobility parameter $\theta_P$ which accounts for both flood and human characteristics. In addiction, a simplified 3D numerical model describing a detailed human geometry partly immersed in water was built to reproduce a selection of the existing experimental data, which is the first example of numerical investigation on the instability conditions of people in floodwater. In general, the topic of this article is appropriate for this journal, and it should be of interest to researchers in the contexts of flood risk analysis and management. However, there exist some obvious errors in the derivation of the dimensionless coefficients $\theta_P$, and the current version can not be accepted, but encourage the authors to re-submit the revised version with major revision.

**Specific Comments:**

(1) The lift force (**Li**) should not be included in the force analysis.

Lift force is generated by pressure difference which results from the flow velocity difference of a completely submerged object. That is to say, the flow velocity at the upper surface of the submerged object is greater than that at the lower surface, so the top pressure is lower than the bottom one (Chien and Wan, 1999). A human body is not completely submerged in floodwaters, and additionally there is little flow water between the bottom surface and the human feet. Therefore it is not necessary to account for the lift force.

Chien N and Wan ZH (1999). Mechanics of sediment transport. ASCE Press, Reston VA.

(2) Such an assumption of $\rho_p = \rho$ is problematic in this derivation.

[The average density of the human body ($\rho_p$=1062 kg/m$^3$) is generally assumed equal to the density of muddy water, thus $\rho_p$ is substituted with $\rho$ in Eq. (2).]

The expression of $\rho_p = \rho$ indicates a human body in floodwater can float, instead of sliding or toppling instability. Therefore it is not appropriate to substitute $\rho_p$ with $\rho$ in Eqs. (6) and (13).

(3) The Drag and lift forces are related to the wetted area rather than the total area of the human body. Therefore in the Eqs. (4) and (5), the wetted area projected to the incoming flow should be equal to **H l** rather than (**H_p l**). Although the wetted water depth does not coincide with the undisturbed water depth **H**, this slight difference can be neglected. So all the following derivation processes are incorrect.

Please see the correct derivation presented as follows:

Sliding:

The sliding instability condition is

$$D > (W - B) \cdot \mu \tag{1}$$

Drag force is a function of the square of flow velocity U and the wetted area *H l*.

$$D = \frac{1}{2} \cdot \rho \cdot U^2 \cdot C_D \cdot H \cdot l \tag{4}$$

$$\frac{1}{2} \cdot \rho \cdot U^2 \cdot C_D \cdot H \cdot l = [\rho_p \cdot g \cdot (H_p \cdot d \cdot l) - \rho \cdot g \cdot (H \cdot d \cdot l)] \cdot \mu \tag{6}$$

$$\frac{U^2 C_D}{\mu g} = \frac{2d(H_p \rho_p - H\rho)}{H\rho} \tag{8}$$

Toppling

The toppling instability condition is

$$(W - B) \cdot d = D \cdot \frac{H}{2} \tag{12}$$

$$[(\rho_p \cdot g \cdot l \cdot H_p \cdot d) - (\rho \cdot g \cdot l \cdot H \cdot d)] \cdot d = (\frac{1}{2} \cdot \rho \cdot U^2 \cdot C_D \cdot H \cdot l) \cdot \frac{H}{2} \tag{13}$$

$$gd^2(H_p - H) = \frac{1}{4}U^2 H^2 \tag{14}$$

(4) The dimensionless mobility parameter $\theta_P$ indicates that the stability degree of a human body in floodwater is only related to the body height and the flow condition. However, all the previous studies show that the stability degree of a human body is related to both body height and body mass (Abt et al., 1989; Jonkman et al., 2008; Xia et al., 2014; and so on). The difference is attributed by the neglect of the difference between $\rho_p$ and $\rho$.

---

## Author Comment (AC2)

First of all, thank you for the interest in our manuscript and for your criticisms. On behalf of my co-authors I will reply point by point to your comments.

(1) "For a stationary body in a moving flow Lift force is the force component acting normal to the mean direction of the undisturbed flow "(Vickery, 1966. Fluctuating lift and drag on a long cylinder of square cross-section in a smooth and in a turbulent stream. Journal of Fluid Mechanics, 25, 03, pp 481-494). Following this definition, the authors think that the hydrodynamic force component, which is directed vertically in our case, cannot be neglected in a full 3D flow around a human body.

Lift force exists also for partly submerged objects as argued by Malavasi and Guadagnini (2003, Hydrodynamic Loading on River Bridges. Journal of Hydraulic Engineering, Vol. 129, No. 11.) who measured drag and lift forces acting on a partly submerged bridge deck through experimental tests. Arslan at al. (2013, Turbulent Flow Around a Semi-Submerged Rectangular Cylinder Journal of Offshore Mechanics and Arctic Engineering, Vol. 135) showed the effect on drag and lift forces of different submergence levels (accounting for partly submerged conditions) on a rectangular cylinder. Arslan et al (2013) used a CFD model to estimate drag and lift forces, which reproduced accurately the experimental data used for validation, where the forces were measured through a dynamometer. Moreover, also Milanesi et al. (2015, A conceptual model of people's vulnerability to floods. Water Resources Research, 51, doi:10.1002/2014WR016172.) in his conceptual model on people's vulnerability to floods accounted for Lift force.

(2)  $\rho_p = \rho$  is a simplifying assumption which is conservative and thus in favour of stability. If we consider  $\rho_p = 1062 \text{ kg/m}^3$  and we do not make any assumption of human body density, we obtain the following mobility parameter

$$\theta_p = \frac{2d}{H_p} \cdot \frac{\frac{\rho_p}{\rho} \cdot H_p - H}{H}$$

It differs from eq.11 for a constant, which is the ratio between the human body density and water density  $\frac{\rho_p}{\rho} = 1.062$ . From the point of view of the

dimensional analysis nothing changes, but the height of the subject appears increased of about the 6%. If the new mobility parameter is calculated for the experimental data the regression curve of Fig. 1 will be shifted, thus it will have a different representative equation. Moreover, for the range of submergence levels and flow velocities tested in the experiments and reproduced numerically in the manuscript, hydrodynamic forces (especially drag force) are more significant the static forces, thus toppling or sliding prevail over floating. This discussion can be added in the final version of the manuscript to clarify the consequences of our hypothesis.

- (3) The authors do not agree with the referee on this point. The selection of the reference area for the hydrodynamic forces is arbitrary, so the use of the wetted area is optional. Drag and lift coefficients in the form of Eqs. 18, 19 are derived from dimensional analysis and the reference area is an arbitrary scale factor with dimensions of (length)2. Thus, wetted area and full frontal area are commonly used in engineering practice, see for instance Fox and McDonald, 1978 (Introduction to Fluid Mechanics, 2nd ed. John Wiley & Sons, N.Y. 684 pp.), Hoerner, 1965 (Fluid dynamic drag, Hoerner Fluid Dynamics ISBN-10: 9993623938), Bertin and Smith, 1979 (Aerodynamics for Engineers, Prentice-Hall, New Jersey, 410pp). Obviously, the choice of the reference area significantly affects the magnitude of the force coefficients but not the forces and the essential is the consistency of the definitions between output of the numerical model and the mobility parameter.
- (4) The dimensionless mobility parameter  $\theta p$  actually indicates that the stability of a human body in floodwaters is related to relative submergence and Froude number. The mass does not appear in the parameter definition because with the dimensional analysis the mass becomes a density  $\rho_p$ . All human subjects tested in the experiments had different mass/weight but had the same density and the dimensional analysis allows identifying dimensionless combinations of the variables of the system for a given set of independent fundamental units. Also if we do not assume  $\rho_p=\rho$  (answer to point 2) we obtain a constant factor 1.062, which virtually increases the height. The height can be also seen as a sort of 'proxy' of the weight for a mesomorphic individual since the mass is the product of body density and body volume (and the body volume depends on the height of the subject).

---

## Author Response (AR1)

**Responses to reviewers**

On behalf of all co-authors I would like to thank the editor and the three referees for their constructive comments, which helped improving the quality of the first version of the manuscript.

Note: the comments raised by the referees are listed with numbers and written in italics, the replies are listed accordingly. Changes to the manuscript are yellow-highlighted.

**Anonymous Referee #1**

*This is a nice paper describing a new hydrodynamic model simulating pedestrians' instability during flooding events. The manuscript includes nice figures and is overall well written, but a double check from a English native speaker would help improve readability.*

The manuscript has been checked to improve readability.

*Yet, I have 2 main major concerns that should be address before publication. (1) The first one is related to the actual usefulness of such a model in risk management. The authors state this point in various parts of the paper, but they do not clarify how such a model can actually be used. I think this clarification is crucial, as its usefulness is one of the selling points of such a model.*

(1R) The main contribution of the work is the identification of the most relevant dimensionless actors playing a role in pedestrians instability. The first and most important one is the relative submergence, which accounts for a characteristic of the subject (height) and for a characteristic of the flow (water depth). The second parameter is Froude number, which is an attribute of the flow. Several authors attributed the scatter of experimental pairs (water depth-velocity) to many variables such as body build, psychological conditions, clothing, training levels etc. The aim of the work is to clarify if a limited number of parameters (i.e. shape factor, submergence, Froude number) is capable of explaining most of the variability observed in the experiments. More generally, dimensional analysis allows merging experiments carried out at various scales, highlighting dependencies and relevant factors. Moving from conventional diagrams (water depth versus velocity) to dimensionless diagrams may help in the definition of safety rules for citizens. This doesn't mean that common people should read a dimensional diagram. As an example, teaching people to recognize a level of submergence with respect to their body (knees, ankles, and waist) is easier than refer to absolute water depths, which are more difficult to assess. These points have been clarified in the introduction of the revised manuscript.

*(2) The second one is related to the lack of a sensitivity analysis. The current modelling exercise does not allow a proper evaluation of the fact that good results are obtained for the right reasons. The paper describes it only with reference to mesh resolution, but there are numerous parameters affecting model outcomes and, by playing with them, one can get a plethora of different results. Thus, to evaluate the proposed model a comprehensive sensitivity analysis is crucial.*

(2R) A sensitivity analysis has been carried out for the dimensionless mobility parameter $\theta_P$ and for the goodness of fit between numerical model and experiments to clarify the role of the main parameters and how they affect the results.

For the mobility parameter $\theta_P$, the sensitivity has been evaluated with respect to: length of the foot $d$, height of the subject $H_p$ and human body density $\rho_P$. The sensitivity is assessed with a local method i.e. calculating the partial derivative of $\theta_P$ with respect to the selected factors $X_j$. A general form of the mobility parameter with $\rho_P \neq \rho$ has been used for the sensitivity analysis and added to the manuscripts (Eq. 12)

$$\theta_P = \frac{2d}{H_P} \cdot \frac{\frac{\rho_p}{\rho} \cdot H_p - H}{H}$$

The following table (Tab. 1 in the revised manuscript) has been added to the text to summarize the outcome of the sensitivity analysis. The analytical formulas allows calculating the sensitivity for each value of the water depth H thus identifying possible critical ranges.

| | Parameter for the sensitivity $X_j$ | | |
|---|---|---|---|
| | $X_j=d$
 *length of the foot* | $X_j=H_p$
 *height of the subject* | $X_j=\rho_P$
 *human body density* |
| Sensitivity function $\frac{\partial \theta_P}{\partial X_j}$ | $\dfrac{2\rho_p}{\rho H} - \dfrac{2}{H_P}$ | $\dfrac{2d}{H_P{}^2}$ | $\dfrac{2d}{\rho H}$ |

The units of measurement of the sensitivity function are length$^{-1}$ and length$^3$/mass for the geometric parameters and density parameter respectively. The sensitivities of $\theta_P$ to $\rho_P$ and $d$ decrease with increasing water depth $H$. The sensitivity to $\rho_P$ and $d$ are of the order of magnitude of $10^{-3}$ and $10^{-1}$-$10^1$ respectively, for the experimental range of water depths. High sensitivity to $d$ is found particularly for water depths lower than 0.5 m. Thus, d is a more sensitive parameter, although its variation is physically constrained because the foot to height ratio is in the range 0.149-0.169 according to allometry studies (Davis, 1990; Pawar and Dadhich, 2012; Fessler et al., 2004). Consequently, small variations of $d$ can be expected. The sensitivity to $H_p$ is of the order of magnitude of $10^{-1}$ for height of the subjects between 1 and 2 m.

Therefore, since the sensitivity of the parameter $\theta_P$ is the product of the sensitivity function and the variation of the parameter, we can conclude that $\theta_P$ is robust enough, although obviously its regression function depends on the experimental data. The above table and comments have been added to section 2.3 of the manuscript.

Regarding the numerical model, a sensitivity analysis to $d$ and friction coefficient $\mu$ has been carried out to understand how the goodness of fit between numerical results and experiments is affected by a change in these parameters. The parameters $d$ and $\mu$ play a role in the calculation of $C_t$ and $C_s$ respectively (Eqs. 16 and 10). A variation of $\pm 10\%$ and $\pm 30\%$ has been applied to both $d$ and $\mu$ one at time and the change in the determination coefficient $R^2$ and RMSE of the fit has been calculated. The results are summarized in Tab. 4, which has been added to the manuscript with the necessary comments (section 4.3).

| Exp. | Human subjects | All data | Human subjects | All data | Human subjects | All data | Human subjects | All data | Human subjects | All data |
|---|---|---|---|---|---|---|---|---|---|---|
| | Base case | | $d$ +10% | | $d$ -10% | | $d$ +30% | | $d$ -30% | |
| $R^2$ | 0.82 | 0.76 | 0.82 | 0.73 | 0.81 | 0.70 | 0.81 | 0.64 | 0.77 | 0.62 |
| RMSE | 0.28 | 0.67 | 0.29 | 0.67 | 0.31 | 0.69 | 0.31 | 0.97 | 0.32 | 1.26 |
| | Base case | | $\mu$ +10% | | $\mu$ -10% | | $\mu$ +30% | | $\mu$ -30% | |
| $R^2$ | 0.82 | 0.76 | 0.82 | 0.72 | 0.82 | 0.74 | 0.78 | 0.69 | 0.80 | 0.71 |
| RMSE | 0.28 | 0.67 | 0.25 | 0.68 | 0.32 | 0.67 | 0.22 | 0.71 | 0.46 | 0.75 |

The results of the sensitivity analysis are overall satisfactory since the determination coefficient $R^2$ does not decrease significantly when modifying the parameters $d$ and $\mu$ both considering only experiments on human subjects both considering all the datasets. In fact, the determination coefficient $R^2$ does not decrease under 0.7. The RMSE tends to increase for larger variations of the parameters especially considering all data, meaning that the scatter plot is sparser but still a high correlation exists between $\theta_P$ and numerical results. A more accurate comparison between numerical results and experiments would be possible if friction and length of the foot were measured during experiments, which is strongly encouraged in future research.

Other variables such as position of the body, posture, clothing etc. play a role in the body-flow hydrodynamic interaction, however the effects of these is left for a future research. The aim of the study in fact, is not the exact reproduction of the laboratory experiments but a better understanding of the overall hydrodynamic force variation on a rigid body with respect to a wide range of flow regimes/submergence levels.

**Anonymous Referee #2**

*The authors introduce an interesting topic that has been addressed by several authors before through an experimental point of view. In the paper there is basically a numerical modelling considering a 3D approach, and some hypothesis associated to it.*

*(1) Some comments of the authors concerning the scatter observed in the experimental studies, must be clarified. Most studies consider different types of individuals to test their problems against urban floods. And it is perfectly logical this scatter, and consider to define hazard criteria as a the lower bound values, not the highest ones. Authors for instance take part of the data of the studied developed by Karvonen, oriented to define hazard criteria, not for the normal people living in the urban area, but for rescue services, people much more trained and strong than aged people or teenagers. So to propose any function including this kind of data, could be biassed not to the normal people. Same comments concerning the message about the tests developed by Foster and Cox, including babies. So it is logical that limiting conditions are not the same for everybody, babies and rescue services, and the scatter must be observed as it is in the real tests. It would be interesting if authors include all the data concerning the previous test developed by other authors (Abt, Takahashi, etc), and their statement that "...scatter can be reduced" considering their parameter is fully right or not.*

(1R)We absolutely agree with the referee on the first point concerning the scatter of existing experimental data. It is perfectly understandable why experiments carried out on different human subjects (i.e. children or professional stuntmen) differ in the corresponding pairs of water depth and velocity. We also stress these 'dimensional' differences in terms of estimated forces while commenting Fig.5 for instance. In the manuscript, we are not arguing that the scatter is not physically

meaningful, but, given the existing scatter, we investigate the hypothesis that a limited number of dimensionless parameters can explain most of the existing variability. In our opinion, the mobility parameter and the dimensional analysis are capable of identifying two main actors in the phenomenon of instability of pedestrians in floodwaters, which are relative submergence and Froude number. That these two 'actors' may be relevant was of course expected by 'scientific common sense', this study not only clearly confirms that, it also estimates to what extent are they relevant and how much is left as unexplained variance. The contribution of these parameters is then investigated through a 3D numerical model. The discussion on the numerical results also include the effect of other parameters, which were not investigated in this study, but are left for a future research.

Regarding the use of the data by Abt et al. and by Takahashi, in the first case we considered the experimental conditions not fully representative of a 3D flow and limited in the investigated range of flow regimes, in the latter it was not possible to retrieve the heights of the tested subjects to calculate the mobility parameter. In order to clarify these points some paragraphs have been added to the manuscript (sections 1 and 2.3).

*(2)Authors have made their choice considering just one orientation, person parallel to the flow. In flood events most of the problems are associated to the crossing of flooding streets. In those cases some of the distances considered in the analysis would be quite different, making more dangerous this condition and not the considered in the paper.*

(2R) It is true that we only considered one flow orientation as a simplifying assumption; also most of the existing experimental studies neglects the effect of the angle of flow incidence. We also considered a rigid body thus we neglected the walking condition. As replied to E. M. Gomariz (SC2 available at http://www.hydrol-earth-syst-sci-discuss.net/hess-2016-261/#discussion), the boundary conditions and set up of the 3D numerical model should be completely modified to account for a non-rigid or moving body and this is out of the scope of the manuscript. Accounting for a walking subject would mean to adopt a fully-coupled CFD-CSD model which would require ad hoc experiments for calibration and validation. Some paragraphs have been added to sect.3.3 to clarify this point.

*(3)Concerning the application of the 3D code, I would like to get some more details. Authors indicate that no turbulence model has been considered. Especially for local effects, turbulence closures are needed to reproduce properly what really happens. The reason why the authors say they do not consider a turbulence model is not clear for me. Results must be checked with a turbulence model and compare if results change or not. Authors calculate few seconds. Is it fully stable flow? Some authors indicate problems when turbulence models are introduced and forces are calculated, and the process leads to some instabilities. Data about CPU time is welcome for the tests made.*

*(3R)*The 3D model aims at investigating the role of the mean flow properties on the phenomenon of instability and a turbulence model was not selected for two main reasons: first, a turbulence model needs the calibration and/or validation of some coefficients and adequate experiments were not available for this purpose; second we are approximating with a rigid body the experimental conditions for subjects which were allowed to move freely. This obviously bears an error in the estimation of the forces on the subject, thus we considered our assumption adequate for the intrinsic uncertainties of the simulated problems. Moreover, as stated in the manuscript, the results of the simplified numerical model were acceptable for a circular cylinder used as a benchmark. The stability of the flow was checked and the parameters were extracted once the simulation reached steady conditions. The flow reaches fast the stability since a wall function is applied (see sect. 3.2) The time required

for simulating 1 s was about 30 minutes with one core and important reductions of the computational time were achieved through parallelization.

The reasons of our hypothesis and the other points have been clarified in the manuscript (sect. 3.1, 3.2)

*(4)Another point not fully clear to me is the consideration of the lift force. Authors make the distinction between the buoyancy and lift force (eq. 1). But next they consider density equals between water and humans. What if not? More physical explanations concerning the final orientation are these forces would be welcome. And more interpretation of the results considering the huge variation of drag and lift coefficients is welcome too. Shape of the "person" tested is the same, so for instance drag coefficient can change with a ratio close to ten, for different rates of submergence. At the beginning seems reasonable, but when the shoes/ foot are covered, increase of submergence affects only to legs, so shape is almost the same. The authors present a variation of the Lift force with Froude number, showing a first decrease, then an increase and another decrease. Can they introduce some physical explanation to this point?*

*(4R)* A general form of the mobility parameter with $\rho p \neq \rho$ (Eq. 12) has been added and a sensitivity analysis has been carried out also on this parameter (section 2.2).

 About the lift force orientation we added a clearer explanation in the text (3.2), however the lift force is oriented downward as long as the only horizontal surface involved in the flow is that of the feet, it becomes upward-oriented when the lower part of the body trunk is reached by the flow.

The interpretation of the variation of drag and lift coefficents has been clarified highlighting in the diagrams (Fig. 4) how the reference area for the force coefficients was initially defined. The increase of drag coefficients with submergence is a consequence of the selection of the whole frontal area of the body as reference area. Fig. 4 has been modified adding the geometric scheme of the human body and the definition of the reference area for the force coefficients. This should resolve the ambiguity of the variation of the forces and force coefficients with the submergence and help in the overall understanding of the manuscript and numerical results.

The physical interpretation of lift force in Fig.5 is the compensation of the effects of submergence and Froude number. The decrease of submergence (linear increase of $C_l$) is counterbalanced by the increase of the square of velocity. This comment has been added to section 4.1.

*(5)Errata: Page 13, line 9, there are two consecutive "this".*

(5R) corrected.

**Anonymous Referee #3**

**General Comment:**

People safety can be compromised when they are exposed to floodwaters that exceed their ability to remain standing, thus leading to a great loss of casualties. Therefore, it is important to study the pedestrian's instability in floodwaters. In order to overcome the scatter of existing experimental data, the authors introduced a dimensionless mobility parameter $\theta P$ which accounts for both flood and human characteristics. In addiction, a simplified 3D numerical model describing a detailed human geometry partly immersed in water was built to reproduce a selection of the existing experimental

data, which is the first example of numerical investigation on the instability conditions of people in floodwater. In general, the topic of this article is appropriate for this journal, and it should be of interest to researchers in the contexts of flood risk analysis and management. However, there exist some obvious errors in the derivation of the dimensionless coefficients $\theta P$, and the current version can not be accepted, but encourage the authors to resubmit the revised version with major revision.

**Specific Comments:**

*(1)The lift force (**Li**) should not be included in the force analysis. Lift force is generated by pressure difference which results from the flow velocity difference of a completely submerged object. That is to say, the flow velocity at the upper surface of the submerged object is greater than that at the lower surface, so the top pressure is lower than the bottom one (Chien and Wan, 1999). A human body is not completely submerged in floodwaters, and additionally there is little flow water between the bottom surface and the human feet. Therefore it is not necessary to account for the lift force. Chien N and Wan ZH (1999). Mechanics of sediment transport. ASCE Press, Reston VA.*

(1R) "For a stationary body in a moving flow Lift force is the force component acting normal to the mean direction of the undisturbed flow "(Vickery, 1966. Fluctuating lift and drag on a long cylinder of square cross-section in a smooth and in a turbulent stream. Journal of Fluid Mechanics, 25, 03, pp 481- 494). Following this definition, the authors think that the hydrodynamic force component, which is directed vertically in our case, cannot be neglected in a full 3D flow around a human body. Lift force exists also for partly submerged objects as argued by Malavasi and Guadagnini (2003, Hydrodynamic Loading on River Bridges. Journal of Hydraulic Engineering, Vol. 129, No. 11.) who measured drag and lift forces acting on a partly submerged bridge deck through experimental tests. Arslan at al. (2013, Turbulent Flow Around a Semi-Submerged Rectangular Cylinder Journal of Offshore Mechanics and Arctic Engineering, Vol. 135) showed the effect on drag and lift forces of different submergence levels (accounting for partly submerged conditions) on a rectangular cylinder. Arslan et al (2013) used a CFD model to estimate drag and lift forces, which reproduced accurately the experimental data used for validation, where the forces were measured through a dynamometer. Moreover, also Milanesi et al. (2015, A conceptual model of people's vulnerability to floods. Water Resources Research, 51, doi:10.1002/2014WR016172.) in his conceptual model on people's vulnerability to floods accounted for lift force. The definition of lift force as in Vickery (1966) has been added to section 2.1.

*(2) The average density of the human body ($\rho p$=1062 kg/m3) is generally assumed equal to the density of muddy water, thus $\rho p$ is substituted with $\rho$ in Eq. (2).] Such an assumption is problematic in this derivation.*
*The expression of $\rho p$ =$\rho$ indicates a human body in floodwater can float, instead of sliding or toppling instability. Therefore it is not appropriate to substitute $\rho p$ with $\rho$ in Eqs. (6) and (13).*

(2R) $\rho_p$=$\rho$ is a simplifying assumption which is conservative and thus in favour of stability. If we consider $\rho_p$=1062 kg/m$^3$ and we do not make any assumption of human body density, we obtain the following mobility parameter which has been added (Eq. 12) to the manuscript

$$\theta_P = \frac{2d}{H_P} \cdot \frac{\frac{\rho_p}{\rho} \cdot H_p - H}{H}$$

It differs from Eq.11 for a constant, which is the ratio between the human body density and water density $\frac{\rho_p}{\rho} = 1.062$ . From the point of view of the dimensional analysis nothing changes, but the height of the subject appears increased of about the 6%. If the new mobility parameter is calculated for the experimental data the regression curve of Fig. 1 will be shifted, thus it will have a different representative equation. Moreover, for the range of submergence levels and flow velocities tested in the experiments and reproduced numerically in the manuscript, hydrodynamic forces (especially drag force) are more significant the static forces, thus toppling or sliding prevail over floating. A sensitivity analysis of $\theta_P$ to $\rho_P$ has been also carried out and added to the text and demonstrated that small variations of $\rho_P$ are negligible because the sensitivity function $\frac{\partial \theta_P}{\partial X_j}$ when $X_j = \rho_P$ has an order of magnitude of $10^{-3}$ (section 2.2, table 1).

*(3) The Drag and lift forces are related to the wetted area rather than the total area of the human body. Therefore in the Eqs. (4) and (5), the wetted area projected to the incoming flow should be equal to $H \cdot l$ rather than ($H_p \cdot l$). Although the wetted water depth does not coincide with the undisturbed water depth $H$, this slight difference can be neglected. So all the following derivation processes are incorrect. Please see the correct derivation presented ... [Equations not reported here].*

(3R)The authors do not agree with the referee on this point. The selection of the reference area for the hydrodynamic forces is a matter of choice among different possibilities. The use of the wetted area is one possibility; the total area is another valid possibility that has been judged as more convenient in the present study. Drag and lift coefficients in the form of Eqs. 19, 20 are derived from dimensional analysis and the reference area is a scale factor with dimensions of (length)$^2$. Thus, wetted area and full frontal area are both commonly used in engineering practice, see for instance Fox and McDonald, 1978 (Introduction to Fluid Mechanics, 2nd ed. John Wiley & Sons, N.Y. 684 pp.), Hoerner, 1965 (Fluid dynamic drag, Hoerner Fluid Dynamics ISBN-10: 9993623938), Bertin and Smith, 1979 (Aerodynamics for Engineers, Prentice-Hall, New Jersey, 410pp). Moreover, the difference between actual water depth and undisturbed water depth is not negligible for supercritical flows as clearly visible in Fig. 3 where actual water depth is almost twice the undisturbed one. Obviously, the choice of the reference area significantly affects the magnitude of the force coefficients but not the forces and the essential is the consistency of the definitions between output of the numerical model and the mobility parameter. In order to clarify this point an explanation has been added to sections 2.2 and 3.2 and Fig. 4 has been modified to show the reference area used for the force coefficients evaluation.

*4) The dimensionless mobility parameter $\theta_P$ indicates that the stability degree of a human body in floodwater is only related to the body height and the flow condition. However, all the previous studies show that the stability degree of a human body is related to both body height and body mass (Abt et al., 1989; Jonkman et al., 2008; Xia et al., 2014; and so on). The difference is attributed by the neglect of the difference between $\rho_p$ and $\rho$.*

(4R)The dimensionless mobility parameter $\theta_p$ actually indicates that the stability of a human body in floodwaters is related to relative submergence and Froude number. The mass does not appear in the parameter definition because with the dimensional analysis the mass becomes a density $\rho_p$. All human subjects tested in the experiments had different mass/weight but had the same density and the dimensional analysis allows identifying dimensionless combinations of the variables of the system for a given set of independent fundamental units. Also if we do not assume $\rho_p=\rho$ (answer to point 2) we obtain a constant factor 1.062, which virtually increases the height. The height can be also seen as a sort of 'proxy' of the weight for a mesomorphic individual since the mass is the product of body density and body volume (and the body volume depends on the height of the subject). According to the sensitivity analysis to $\rho_p$ the relevance of the body density would be more significant for very small water depths, which require very high velocity to compromise pedestrians' stability. In these conditions sliding instability would prevail, with important role of weight and friction coefficient. However, the influence of body type and build on the occurrence of instability could be a topic on its own for a future research. These comments have been added to section 2.2 to clarify this point.

[revised manuscript text omitted]

---

## Author Response (AR2)

**Responses to reviewers**

Comments raised by the referees are listed with numbers and written in italics, the replies are listed accordingly. Changes to the manuscript are yellow-highlighted

**Anonymous Referee 1**

Anonymous Referee 1 suggested publication without changes.

**Anonymous Referee 2**

*(1) The authors have addressed some of the point remarked in the previous revision, but still there are some minor points concerning the paper. The justification of the lift force is not fully done, so in the same line as the other reviewer would like more details on it.*

(1R) Conventionally lift force is considered upward directed since most of the literature refers to aerodynamic/hydrodynamic investigations on fully submerged bodies. However, the general definition of lift action is the force component acting normally to the mean undisturbed flow direction (Vickery, 1966). In fact, experimental and numerical studies suggest that lift force does exist for partly submerged bodies (Malavasi and Guadagnini, 2003) and can be upward and downward directed (Arslan et al., 2013). The human body has a complex shape and its hydrodynamic interaction is affected not only by the flow but also by the portions of the body involved. For the analysed range of water depths, three parts can be distinguished: feet, legs and trunk. Since the lift force is the integral of pressures on the surface, its value is significantly affected by submergence. In fact, for low water depths, legs only contribute to drag force and feet are subject to a vertical force downward directed, given the assumption of adherence between bottom and feet. Once the pelvis is wetted, and this may occur for undisturbed water depth lower than body trunk due to backwater effects, an upward directed action is added to the downward directed feet action (conventionally negative). With the increase of submergence, the upward component increases until the global vertical force becomes fully positive and this explains the lift force behaviour in Fig. 5.

The above clarifications have been added to sections 4.1 (Results) and 5 (Discussion).

*(2) The comments concerning the different type of people participating in the previous studies is addressed, but still authors pretend to eliminate the scatter observed in previous studies so this is not possible.*

(2R) Conceptual models on people instability developed so far usually refer to rigid mechanisms (i.e. toppling and sliding). The scatter of experimental data represented as critical pairs of water depth and velocity is widely highlighted as a drawback limiting the implementation of physically-based hazard curves. This work demonstrate that a 'rigid' conceptual model adopted in a dimensionless space is capable of explaining most of the observed experimental variability. The scatter is strongly reduced, not eliminated. Unexplained variance remains but we identify in the height of the subject ($H_P$) the most relevant anatomic characteristic to understand the occurrence of instability. Moreover, dimensional differences between the subjects are stressed (see comments to Fig. 5)

This clarification has been added to the conclusions (section 6).

*(3) To reduce the stability process to a simple formula is suggesting but authors who worked in previous year consider always a limiting value or curve bounding all the experimental data.*

(3R) The $\theta_{Pcr}$ curve is, by construction, a mechanical equilibrium threshold, which is meaningful for $0 < H < H_P$. From the physical point of view H=0 corresponds to extremely high Fr tending to infinity, $H = H_P$

corresponds to fully submerged condition where static equilibrium occurs given the assumption $\rho_P=\rho$. For practical applications and risk mapping a safety factor could be applied to shift down the threshold curve and account for model uncertainties and experimental variance.

These comments have been added to section 2.2 and 5.

(4) *The comments about the use of a parameter based on Froude is again interesting but it is not clear if this is better than traditional vy functions. Froude is a dimensionless parameter and the hyperbolic form vy=a is not, but aside this I see no advantages, honestly. A function equal for everybody just as a function of Fr number, not considering the differences between different types of persons, seems difficult to accept. Hyperbolic curves can distinguish with the "a" value of the vy=a in front of different individuals.*

The critical threshold curve $\theta_{Pcr}$ is unique and depends on Froude number but this does not imply that differences between subjects are not accounted for. In fact, the assessment of the human body stability relies on the comparison between the value of the mobility parameter $\theta_P$ and the critical threshold $\theta_{Pcr}$ for a given Fr. Thus, when $\theta_P$ is calculated, the height of the subject is required (see Eq. 11 and 17). The diagram in Fig. 8, where different stable/unstable portions are identified, helps explaining the use of the dimensionless approach for practical applications. Let's assume a flood model returns H=0.5 m and U=1.1 m/s (i.e. Fr= 0.5). Let's consider two men 1.95 m and 1.6 m tall respectively. The critical threshold $\theta_{Pcr}$ for Fr=0.5 is 0.84 (Eq. 18). For the first man $\theta_P =0.87$ ($\theta_P> \theta_{Pcr}$ so stability is expected), for the other $\theta_P =0.66$ ($\theta_P< \theta_{Pcr}$ so instability is expected). This means that our criterion is able to distinguish different individuals. On the contrary, in the traditional vy functions a distinction between different subject characteristics is not possible. Different values of "a" in the curves vy=a do not necessarily correspond to clearly identifiable (and measured) subject peculiarities since the curves are empirical, they do not account for subject characteristics and are not physically based.

This clarification has been added to section 5 and 6.

[revised manuscript text omitted]